# Evidence and magnitude of the effects of meteorological changes on SARS-CoV-2 transmission

**Adam Kaplin**[1¤]*, **Caesar Junker**[2], **Anupama Kumar**[1], **Mary Anne Ribeiro**[3], **Eileen Yu**[1,4], **Michael Wang**[1], **Ted Smith**[5], **Shesh N. Rai**[5,6], **Aruni Bhatnagar**[5]

**1** Departments of Psychiatry & Neurology, Johns Hopkins University SOM, Baltimore, Maryland, United States of America, **2** Joint Artificial Intelligence Center, Pentagon, Washington, DC, United States of America, **3** PUPA, Sao Paulo, Brazil, **4** Case Western Reserve University School of Medicine, Cleveland, Ohio, United States of America, **5** Christina Lee Brown Envirome Institute, University of Louisville School of Medicine, Louisville, Kentucky, United States of America, **6** Department of Bioinformatics and Biostatistics, University of Louisville, Kentucky, United States of America

¤ Current address: MyMD Pharmaceuticals, Inc., Baltimore, Maryland, United States of America
* aikaplin@mymd.com

**Data Availability Statement:** The manuscript describes where all data was obtained, and all data is publicly available (see links below). We have also included an URL to excel files where all of the data

# Abstract

## Importance

Intensity and duration of the COVID-19 pandemic, and planning required to balance concerns of saving lives and avoiding economic collapse, could depend significantly on whether SARS-CoV-2 transmission is sensitive to seasonal changes.

## Objective

Hypothesis is that increasing temperature results in reduced SARS CoV-2 transmission and may help slow the increase of cases over time.

## Setting

Fifty representative Northern Hemisphere countries meeting specific criteria had sufficient COVID-19 case and meteorological data for analysis.

## Methods

Regression was used to find the relationship between the log of number of COVID-19 cases and temperature over time in 50 representative countries. To summarize the day-day variability, and reduce dimensionality, we selected a robust measure, Coefficient of Time (CT), for each location. The resulting regression coefficients were then used in a multivariable regression against meteorological, country-level and demographic covariates.

## Results

Median minimum daily temperature showed the strongest correlation with the reciprocal of CT (which can be considered as a rate associated with doubling time) for confirmed cases (adjusted $R^2$ = 0.610, p = 1.45E-06). A similar correlation was found using median daily

used in our calculations is available in excel sheets that are labelled for ease of use: The URL is https://www.dropbox.com/sh/d32xjzy1luifty2/AABylbaD985DyLRmKGEGLxoWa?dl=0 Data Repositories Referenced in Manuscript: Meteorological Data: https://www7.ncdc.noaa.gov/CDO/cdoselect.cmd. Novel Coronavirus (COVID-19) Cases, provided by JHU CSSE https://github.com/CSSEGISandData/COVID-19 Land area for each country was obtained from Worldometer (https://www.worldometers.info/coronavirus/#countries) and median age from CIA World Factbook (https://www.cia.gov/the-world-factbook/field/median-age/).

**Funding:** The authors received no specific funding for this work. MAAR is an employee of Pupa but contributed to this work independently of this affiliation. Pupa had no role in study design, data collection and analysis, decision to publish, or preparation of the manuscript.

**Competing interests:** The authors of the study have read the journal's policy, and have the following competing interests to declare: MAAR is an employee of Pupa but contributed to this work independently of this affiliation. AK is currently employed by MyMD Pharmaceuticals; however, this affiliation was not held at the time this study was conducted. This does not alter our adherence to PLOS ONE policies on sharing data and materials. There are no patents, products in development or marketed products associated with this research to declare. The authors have declared that no other competing interests exist.

dewpoint, which was highly colinear with temperature, and therefore was not used in the analysis. The correlation between minimum median temperature and the rate of increase of the log of confirmed cases was 47% and 45% greater than for cases of death and recovered cases of COVID-19, respectively. This suggests the primary influence of temperature is on SARS-CoV-2 transmission more than COVID-19 morbidity. Based on the correlation between temperature and the rate of increase in COVID-19, it can be estimated that, between the range of 30 to 100 degrees Fahrenheit, a one degree increase is associated with a 1% decrease—and a one degree decrease could be associated with a 3.7% increase—in the rate of increase of the log of daily confirmed cases. This model of the effect of decreasing temperatures can only be verified over time as the pandemic proceeds through colder months.

## Conclusions

The results suggest that boreal summer months are associated with slower rates of COVID-19 transmission, consistent with the behavior of a seasonal respiratory virus. Knowledge of COVID-19 seasonality could prove useful in local planning for phased reductions social interventions and help to prepare for the timing of possible pandemic resurgence during cooler months.

## Introduction

Severe Acute Respiratory Syndrome Coronavirus 2 (SARS-CoV-2), is a newly-identified enveloped, non-segmented, positive sense RNA virus. It is responsible for Coronavirus Disease 2019 (COVID-2019), which was designated a pandemic by the World Health Organization (WHO) on March 11, 2020 [1]. SARS-CoV-2 belongs to a large family of human corona viruses such as MERS-CoV and SARS-CoV-1, which are respiratory pathogens associated with a range of respiratory and non-respiratory outcomes. Like other respiratory infectious agents, most coronaviruses display marked seasonality [2]. Seasonality in respiratory viruses is characterized by increased transmission in cooler, less humid months and decreased transmission in warmer and more humid months. It has been demonstrated in the laboratory that for most respiratory viruses, including Influenza, SARS-CoV-1 and SARS-CoV-2, the stability of the envelope surrounding and protecting these viruses is temperature sensitive—degrading more rapidly at higher temperatures and showing increased stability at cooler temperatures [3]. In addition to temperature, the transmission of respiratory viruses demonstrating seasonality are also impacted by humidity, where lower humidity leads to smaller droplets that are more volatile and capable of lodging deeper in the lung [3]. Usually, rather than humidity, dewpoint is substituted instead in meteorological reports. Dew point is the temperature, at constant pressure, to which the air needs to be cooled in order to achieve a 100% relative humidity [4]. At that point the air cannot hold more water in the gas form. SARS-CoV-1, which shows seasonal changes with decreased rates of transmission with elevated temperatures, is genetically 80% identical to SARS-CoV-2 [4, 5].

Because SAR-CoV-2 is a recently identified human pathogen, seasonal variations in its transmission have not become evident, though there is much speculation that seasonal change with the boreal summer might decrease infection rates of SARS-CoV-2 and help flatten the epidemic curve of COVID-19 [6]. Several studies, varying in their geographical sampling,

methodology and findings, have endeavored to shed light on this subject. With respect to geography, four out of seven of the peer-reviewed studies that attempted to establish whether SARS-CoV-2 has seasonal changes in its transmissibility examined rates in China [7–10], two others restricted themselves to either Japan or France [11, 12], and two examined a collection of cities or countries around the world [7, 13]. Methodologies included the use of generalized additive model, poisson regression analysis, and univariate and multiple regression modeling [7, 9–11, 14]. The findings concerning whether SARS-CoV-2 is likely to display seasonality in its rates of transmission varied widely, including: a) two studies reported an asymmetric impact of temperature (one being biphasic at 50˚F, and another demonstrating effects only below 37.4˚F [8, 15], b), one showed no influence of temperature on COVID-19 rates [9], c) two showed inconclusive results with respect to the likelihood of seasonality of SARS-CoV-2 [8, 13], d) one showed distribution of substantial community outbreaks of COVID-19 along restricted latitude, temperature, and humidity measurement [13] and e) one showed a sensitivity to temperature that was of such large magnitude (i.e. a 50% decrease in rate of transmission with every 1 degree increase in average temperature) that it seemed incongruous with observed variations across the globe [14].

A better understanding of the seasonal dependence of SARS-CoV-2 infections, and the identification of environmental conditions that regulate its spread, would be helpful in assessing the differential impact of the virus in different geographic locations under various meteorological conditions. Such knowledge would also help in predicting the timing and degree of potential slowing of transmission in the summer and resurgence of the virus during winter [16].

To understand the seasonal dependence of SARS-CoV-2 infections in the absence of year-long incidence data, we examined the effect of atmospheric temperature, the main variable with season [16], on SARS CoV-2 transmissibility using currently available data on the rates of SARS-CoV-2 infection during winter and spring from fifty representative countries in the Northern Hemisphere.

## Methods

### Geographical sampling criteria

Geographical changes in cumulative COVID-19 confirmed deaths, and recovered cases from 1/22/20 through 4/6/20 were estimated from databases made available by the Johns Hopkins University [17]. The start date of January 22, 2020 was chosen because that is when the database started tracking cases. For analysis, we selected only the Northern Hemisphere countries because even though the number of cases have been increasing on both sides of the equator, only the Northern Hemisphere experienced an increase in temperature during the study period. The opposite was true in the Southern Hemisphere. As a result, the impact of temperature on the rate of new cases cannot be investigated simultaneously on pooled data from both hemispheres. We excluded tropical regions because of the minimal change in temperature throughout the year. Italy was excluded because of its "massive underreporting and undertesting of COVID-19," with an estimate of only 2% of total cases being reported based on calculations involving crude case fatality risks [18].

We also excluded from analysis China [19] and Russia [20]. We excluded China and other countries with multiple data sites (for both COVID-19 cases and NOAA meteorological data) within its borders. Each country had differing responses to the increasing rates of COVID-19 within its borders (e.g. the timing and availability of testing, government policy decisions about when and how robustly to respond with non-pharmaceutical interventions, etc). Because each country that was included contributed data representing a single response to

COVID-19, the variability in the response would be randomized (some countries more and some less in various aspects of their response). The inclusion of a single country with numerous measures within its borders would risk creating a systematic bias for that country in terms of a potential effect on the rate of transmission of COVID-19. China had 33 provinces for which cases were collected from JHU COVID-19 Data Repository by the Center for Systems Science and Engineering (CSSE) at Johns Hopkins University (https://github.com/CSSEGISandData/COVID-19). Thus, to preserve methodological conformity, countries like China with separate reports of cases from multiple regions were excluded.

There is also an abundance evidence to support the dramatic under-reporting of cases from China before the time the JHU COVID-19 Data Repository database begins on 1/22/20, thus suggesting that trying to base results on non-JHU sources prior to that date would be fraught with large inaccuracies. For example, a recent publication by Krantz et al [21] calculated the ratio of reported COVID-19 cases and contrasted them with model-based predictions of COVID-19 for 8 major countries. The ratio of reported to infected cases for China was between 1:149 to 1:1104. In comparison, France was 1:5 and Germany was 1:4. Based on cremation related information from China, He et al [22] were able to calculate COVID-19 case estimates projected on February 7, 2020 in Wuhan ranged from 305,000 to 1,272,000 for infections that were at least 10 times the official figure of 13,603. The same source of information implied starting time of the outbreak is October 2019, rather than the government reported date of 12/31/19 [22].

Including Russia presented a problem of size. A country of Russia's magnitude had only a single JHU CSSE COVID-19 data entry with a single latitude and longitude, which could clearly not accurately represent the appropriate temperature corresponding to the rate of increase of cases for the entire country. Russia has the largest land area of any country in the world, with vastly differing temperatures across its vast expanse, being comprised of 17 million square kilometers. The largest country included in the 50 analyzed in our manuscript was Algeria, that has a land area of 2.4 million square kilometers, which is 7-fold smaller than Russia. One temperature for such a vast country with a huge range of local climates would not yield accurate data. Thus, Russia was excluded.

Other countries were excluded due to limited access to, or the lack of, reliable data on COVID-19 cases or meteorological data. To enable analysis of cases during consecutive days in a log linear fashion, we established the following criteria to ensure sufficient numbers of confirmed cases: to be included, during time period 1/22/20-4/6/20, countries needed to have a median number of cumulative confirmed cases >1 or the mean number of cumulative confirmed cases >75.

The resulting 50 representative countries (Table 1) had a mean latitude of 38 +/-15 SD, range 1–65 degrees, that covers 70% of the total range (1–90 degrees), with each country being separated from the next by a mean of 1.3 +/-1.7 SD degrees. Countries had a mean longitude of 34+/-44 SD, range -103 to 138, that covered 70% of the total range (-180 to 180 degrees), with each country separated from the next by a mean of 6 +/- 13 SD degrees. The resulting wide range of temperatures across the 50 countries was as follows: 1) mean average temperature 51.1 +/-16.9˚F, median 44.9˚F, range 22.6–85.7˚F, and 2) mean minimal temperature 42.0 +/-16.3˚F, median 36.6˚F, range 15.5–77.4˚F.

## Meteorological variables

Daily weather data for each of the 50 countries were obtained from National Oceanic and Atmospheric Administration (NOAA) [23]. Daily minimum temperatures (Tmin) were used because they appear to be more important for virus transmissibility [24]. We used Dew Point

**Table 1. First regression: Rate of rise of confirmed cases of COVID-19 in 50 countries, and the associated local meteorological and demographic variables.**

| Country | CT Conf Cases | CT$^{-1}$ Conf Cases | $R^2$ | Tmin Median | DP Median | Land Area Per Capita | Median age | Conf Cases Days |
|---|---|---|---|---|---|---|---|---|
| Algeria | 0.081 | 12.42 | 0.969 | 56.3 | 25.2 | 0.05431 | 28.9 | 42.0 |
| Armenia | 0.103 | 9.74 | 0.909 | 26.6 | 25.2 | 0.00961 | 36.6 | 37.0 |
| Austria | 0.097 | 10.26 | 0.941 | 29.1 | 31.2 | 0.00915 | 44.5 | 42.0 |
| Belgium | 0.090 | 11.17 | 0.913 | 38.7 | 37.4 | 0.00261 | 41.6 | 63.0 |
| Bosnia and Herzegovina | 0.086 | 11.68 | 0.967 | 20.3 | 23.8 | 0.01554 | 43.3 | 33.0 |
| Croatia | 0.074 | 13.59 | 0.977 | 32.9 | 32.8 | 0.01363 | 43.9 | 42.0 |
| Czechia | 0.091 | 10.93 | 0.942 | 33.0 | 30.2 | 0.00721 | 43.3 | 37.0 |
| Egypt | 0.076 | 13.21 | 0.928 | 48.3 | 38.7 | 0.00973 | 24.1 | 53.0 |
| Estonia | 0.088 | 11.36 | 0.908 | 31.5 | 32.5 | 0.03196 | 43.7 | 40.0 |
| Finland | 0.062 | 16.13 | 0.899 | 19.3 | 22.3 | 0.05485 | 42.8 | 69.0 |
| France | 0.071 | 14.03 | 0.940 | 34.6 | 38.2 | 0.00839 | 41.7 | 74.0 |
| Germany | 0.074 | 13.49 | 0.940 | 34.7 | 33.4 | 0.00416 | 47.8 | 71.0 |
| Greece | 0.074 | 13.52 | 0.904 | 40.1 | 42.2 | 0.01237 | 45.3 | 41.0 |
| Hungary | 0.081 | 12.29 | 0.965 | 32.0 | 31.3 | 0.00937 | 43.6 | 34.0 |
| Iceland | 0.072 | 13.81 | 0.881 | 15.5 | 17.4 | 0.29378 | 37.1 | 39.0 |
| India | 0.055 | 18.32 | 0.903 | 64.0 | 56.6 | 0.00215 | 28.7 | 68.0 |
| Iran | 0.076 | 13.18 | 0.820 | 34.8 | 18.5 | 0.01939 | 31.7 | 48.0 |
| Iraq | 0.058 | 17.13 | 0.913 | 49.3 | 45.5 | 0.01080 | 21.2 | 43.0 |
| Ireland | 0.104 | 9.66 | 0.947 | 35.8 | 37.4 | 0.01395 | 37.8 | 38.0 |
| Israel | 0.096 | 10.45 | 0.986 | 46.0 | 42.3 | 0.00250 | 30.4 | 46.0 |
| Japan | 0.041 | 24.67 | 0.967 | 44.6 | 38.2 | 0.00288 | 48.6 | 76.0 |
| Kuwait | 0.035 | 28.56 | 0.792 | 56.9 | 46.9 | 0.00417 | 43.2 | 43.0 |
| Lebanon | 0.067 | 15.02 | 0.929 | 41.8 | 40.4 | 0.00150 | 29.7 | 46.0 |
| Malaysia | 0.044 | 22.77 | 0.932 | 74.8 | 75.8 | 0.01015 | 33.7 | 73.0 |
| Mexico | 0.089 | 11.23 | 0.967 | 46.4 | 36.7 | 0.01508 | 29.2 | 39.0 |
| Moldova | 0.098 | 10.23 | 0.947 | 31.6 | 29.3 | 0.00814 | 29.3 | 30.0 |
| Morocco | 0.098 | 10.19 | 0.974 | 44.2 | 18.5 | 0.01209 | 37.7 | 36.0 |
| Netherlands | 0.097 | 10.34 | 0.894 | 36.8 | 37.5 | 0.00197 | 29.1 | 40.0 |
| Norway | 0.080 | 12.50 | 0.840 | 21.1 | 21.5 | 0.06738 | 42.8 | 41.0 |
| Oman | 0.049 | 20.28 | 0.972 | 57.9 | 52.9 | 0.06061 | 39.5 | 43.0 |
| Pakistan | 0.095 | 10.51 | 0.952 | 49.6 | 36.5 | 0.00349 | 26.2 | 41.0 |
| Philippines | 0.055 | 18.09 | 0.863 | 74.2 | 76.1 | 0.00272 | 30.1 | 68.0 |
| Poland | 0.102 | 9.80 | 0.923 | 31.1 | 30.9 | 0.00809 | 24.1 | 34.0 |
| Portugal | 0.109 | 9.16 | 0.957 | 47.9 | 48.6 | 0.00898 | 41.9 | 36.0 |
| Qatar | 0.056 | 17.82 | 0.798 | 57.9 | 49.5 | 0.00403 | 33.7 | 38.0 |
| Romania | 0.095 | 10.50 | 0.977 | 28.4 | 30.3 | 0.01196 | 44.6 | 41.0 |
| Saudi Arabia | 0.098 | 10.22 | 0.927 | 59.0 | 29.0 | 0.06175 | 42.5 | 36.0 |
| Serbia | 0.105 | 9.53 | 0.916 | 34.2 | 32.9 | 0.01001 | 30.8 | 32.0 |
| Singapore | 0.031 | 32.04 | 0.932 | 77.4 | 74.7 | 0.00012 | 43.4 | 75.0 |
| Slovakia | 0.079 | 12.65 | 0.841 | 26.5 | 27.4 | 0.00881 | 35.6 | 32.0 |
| Slovenia | 0.067 | 14.83 | 0.803 | 32.3 | 33.3 | 0.00969 | 44.9 | 33.0 |
| South Korea | 0.058 | 17.10 | 0.882 | 29.4 | 23.6 | 0.00190 | 44.9 | 76.0 |
| Spain | 0.100 | 10.01 | 0.949 | 42.2 | 43.0 | 0.01067 | 43.9 | 66.0 |
| Sweden | 0.076 | 13.14 | 0.915 | 18.9 | 22.8 | 0.04063 | 41.1 | 67.0 |
| Switzerland | 0.096 | 10.43 | 0.899 | 30.7 | 33.6 | 0.00457 | 42.7 | 42.0 |
| Thailand | 0.032 | 31.33 | 0.869 | 73.4 | 71.3 | 0.00732 | 39.0 | 76.0 |
| Turkey | 0.177 | 5.65 | 0.924 | 27.4 | 31.1 | 0.00913 | 32.2 | 27.0 |

*(Continued)*

**Table 1.** (Continued)

| Country | CT Conf Cases | CT$^{-1}$ Conf Cases | R$^2$ | Tmin Median | DP Median | Land Area Per Capita | Median age | Conf Cases Days |
|---|---|---|---|---|---|---|---|---|
| United Arab Emirates | 0.039 | 25.87 | 0.950 | 64.8 | 56.9 | 0.00845 | 38.4 | 69.0 |
| United Kingdom | 0.075 | 13.33 | 0.964 | 36.1 | 36.3 | 0.00356 | 40.6 | 67.0 |
| Vietnam | 0.027 | 36.94 | 0.924 | 73.2 | 71.4 | 0.00319 | 31.9 | 75.0 |

Fifty representative countries meeting specific criteria (e.g. all Northern Hemisphere, because seasons in Southern Hemisphere are opposite) are shown. Meteorological data (daily minimum and average temperatures, daily dewpoint) were obtained from NOAA stations closest to the latitude and longitude listed in the JHU COVID-19 database (https://github.com/CSSEGISandData/COVID-19) as the location from which cases in each country were associated. Median minimum temperature (Tmin MEDIAN) and dewpoint (DP MEDIAN) were ascertained for each country over the timeframe that cases were increasing between 1/22/20 (when data began to be collected) and 4/6/20. Median average temperature is included as a reference to provide the reader with additional climate information about each country. Linear regression of the log of the daily cumulative COVID-19 confirmed cases versus time (sequential days) was conducted, starting with the date of the first case for each country, and the resulting regression coefficients (which were termed coefficients of time (CT)) for each country was thereby obtained. R$^2$ are shown for this linear regression in each country. The reciprocal of CT (CT$^{-1}$) was used in a second regression analysis of CT$^{-1}$ vs Tmin MEDIAN, to explore the correlation between temperature and rates of increase in the log of confirmed cases. The second regression included with temperature or dewpoint the additional independent variables of Land Area per capita, Median Age, and Days of Cases for each country, because of the potential influence of these variables on interactions and behavior that could influence rates of transmission. Land area for each country was obtained from Worldometer (https://www.worldometers.info/coronavirus/#countries) and median age from CIA World Factbook (https://www.cia.gov/library/publications/the-world-factbook/fields/343rank.html).

(DP) as a measure of humidity because it is an absolute assessment of water vapor content that does not depend upon temperature, and it is the only country meteorological data related to humidity that is tracked by NOAA. The daily Tmin and DP for each country were obtained from NOAA station locations closest to the specific latitude and longitude that the JHU database listed for each country. The medians of the daily Tmin and daily DP were derived for each country for whatever portion of 1/22/20 through 4/6/20 that they had cases of confirmed COVID-19 reported (Table 1). Median was chosen because it is less affected by outliers and skewed data than the mean.

## Software packages used in analyses

Statistical analysis was performed with StatPlus software for the Mac [25]. The graphing software employed was GraphPad Prism 8 for the Mac [26].

## Corrections applied to account for multiple comparisons

The unit variable was country. At the unit level, dependent and independent variables were evaluated. There were three correlated outcome measures: confirmed, deaths, or recovered cases. For each of these three outcomes, an aggregate measure CT or CT$^{-1}$ was calculated and then associated with multiple independent variables. Due to evaluating three outcomes in the same population, a multiple adjustment was required. Two simple and conservative approaches that have been used include the following: 1. to compare observed p values (p) with adjusted alpha (= alpha/3) or 2. inflate the observed p values by 3 (p$^*$3) and compare at fixed alpha level [27, 28] We adopted the latter approach. Also note that because CT and CT$^{-1}$ are one-to-one transformations, there was no need to adjust for using CT or CT$^{-1}$ as an dependent variable.

## Relationship of CT to doubling time (Td)

To clarify the relative magnitude of our findings, we explored the relationship between CT and doubling time ($T_d$), a standard measure of rate of viral transmission [29]. Doubling times are the intervals at which the cumulative incidence of cases doubles. Under conditions of

exponential, with a constant growth rate r, the doubling time equals (ln 2)/r [30]. Thus, an increase in the doubling time represents a decreased rate of transmission.

To calculate $T_d$ from our data obtained in our first regression (see Results section for details), we took the following steps: to begin, we used our first regression of log of confirmed cases vs time to interpolate on which days the number of cases doubled for each country, utilizing a centered second order polynomial model to fit the data (mean $R^2$ = 0.958, Standard Deviation = +/- 0.025). Because we were using log of confirmed cases, we converted doubling of cases to log form (e.g. 2, 4, 8, 16, 32, 64 corresponds to 0.301, 0.602, 0.903, 1.204, 1.505, 1.806). Since the period of time we measured cases and temperature was 75 days long (1/22/20-4/6/20), we included doubling times up to day 64. We next calculated the average $T_d$ (i.e. number of days) between each interval corresponding to a doubling of confirmed cases. We then performed a linear regression on $T_d$ and CT, and $T_d$ and Tmin Med, with the following results:

## Results

### Analysis of the impact of changing temperature on the rate of rise of COVID-19 cases

To investigate the relationship between Tmin and COVID-19 transmissibility, we devised a two-step process. In the first step, starting with the date of the first case for each country we examined linear regressions of the log of the daily cumulative COVID-19 cases versus time in sequential days (Fig 1). From this analysis we calculated the resulting regression coefficients (which we termed coefficients of time (CT) with units of log cases/day) for each country (Table 1). To determine consecutive days, we converted calendar dates to sequential serial numbers that can be used in calculations—e.g. 1/22/2020 was converted to 43852 because it is 43,851 days after 1/1/1900. The CT of log confirmed cases vs sequential days for all fifty countries yielded a mean overall $R^2$ of 0.919 +/- 0.049 SD, range 0.792–0.986, and p-value mean $4.20 \times 10^{-14} \pm 2.61 \times 10^{-13}$, range $1.00 \times 10^{-15}–1.84 \times 10^{-12}$ (Table 1).

In the second step, we investigated whether CT was associated with the median daily $T_{min}$ (Tmin_Med) in each country for the same time period that cases had been sampled. Because of the approximate proportionality of fitted CT with the standardized residuals when performing linear regression with Tmin_Med, we used a reciprocal transformation of the dependent variable [31] Thus, for the second regression for all countries we used the reciprocal of CT ($CT^{-1}$) (Fig 2).

To account for susceptibility variation resulting from age differences that could affect both the comorbidity levels as well as social gathering behavior, we used each country's population median age (MA) as another covariate [32]. Because the 50 countries varied widely in size, and the density of the population could impact transmission, we used land area per capita (LAPC) as an additional covariate [33]. Finally, each country differed with respect to when it reported its identified cases, so that between the target dates of 1/22/20-4/6/20, there were different durations of time over which the course of transmission occurred. To account for this, the total number of days for each country during which they had cases of COVID-19 between 1/22/20 and 4/6/20 (Days of Cases) was used as a final covariate. In multivariable regression analysis of $CT^{-1}$ versus the median of Tmin_Med (Table 2), we found that the rate of increase of the log of confirmed cases was significantly associated with the covariates Tmin_Med, LAPC, MA and Days of Cases (adjusted $R^2$ = 0.610, p = $1.45 \times 10^{-6}$; see Fig 2 & Table 2).

### Sensitivity of the correlation between $CT^{-1}$ and Tmin_Med

In our two-step model, first we aggregated the outcome measure CT or $CT^{-1}$ at the country level and applied a standard linear regression; second, using multivariable regression analysis

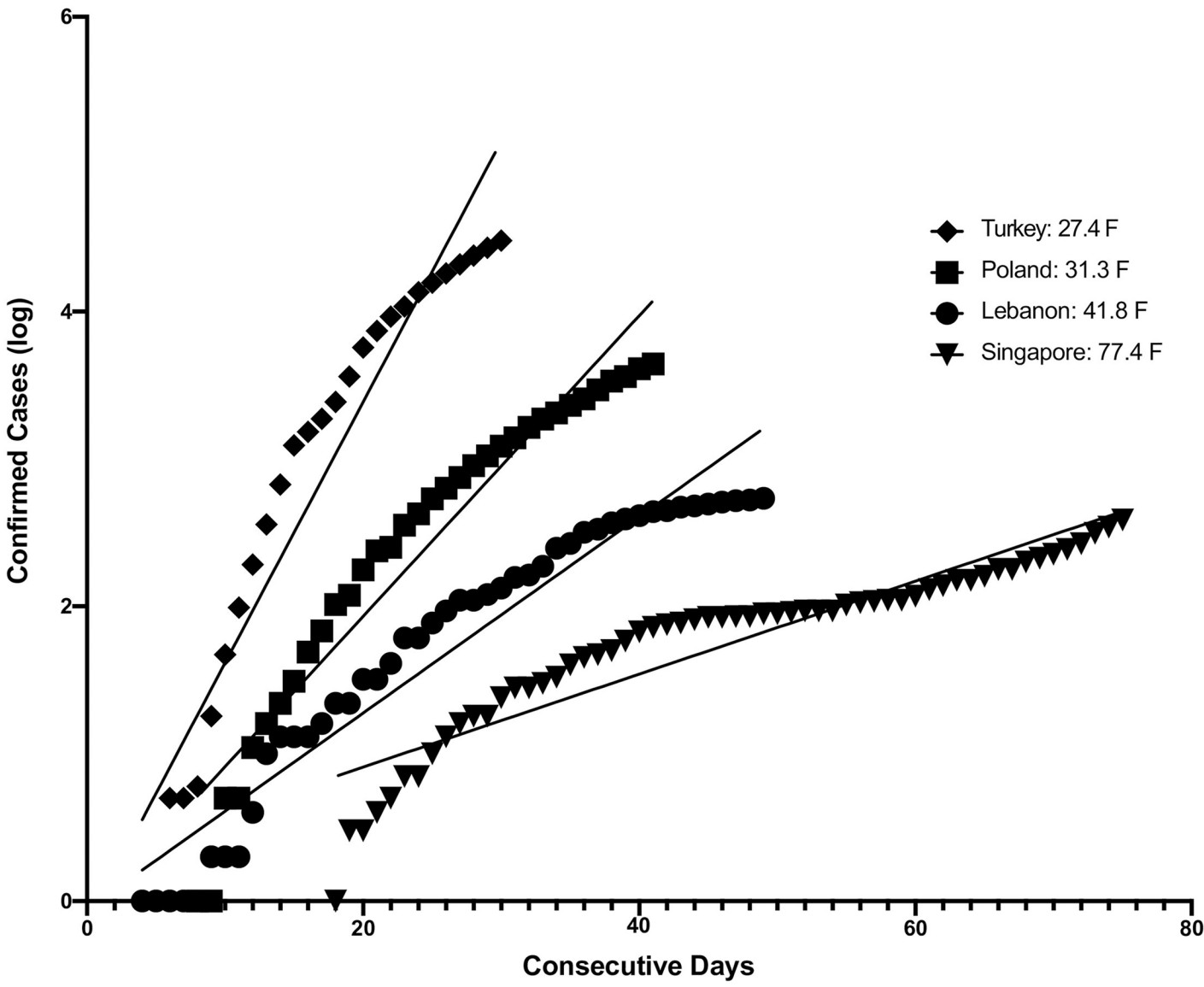

**Fig 1. The first regression of log of confirmed cases vs consecutive days for 4 of the total of 50 countries is shown.** Starting with the date of the first case for each country, which was standardized to day 1 for purposes of visual comparison, we examined linear regressions of the log of the daily cumulative COVID-19 cases versus time in sequential days. From this analysis we calculated the resulting regression coefficients (which we termed coefficients of time (CT) with units of log cases/day) for each country.

we associated CT or $CT^{-1}$ with local meteorological and demographic independent variables. Since the initial aggregation might have played a role in determining response measures, we performed a sensitivity analysis by reducing sampling systematically (by approximately 10–20%) in the first stage [34]. This permitted us to test the robustness of the findings of our model.

The initial selection of days sampled during the 75 day period—1/22/20-4/6/20—was varied by progressively removing 8 or 16 days (11% or 22%) from either the 1/22/20 or the 4/6/20 end, and the resulting periods were run through our two step regression model. The temperature sampling was changed according to the new time periods (i.e. 1/22/20-3/29/20, 1/22/20-3/21/20, 2/7/20-4/6/20, 2/24/20-4/6/20). As shown in S1 and S2 Figs, removal of 8 or 16 days

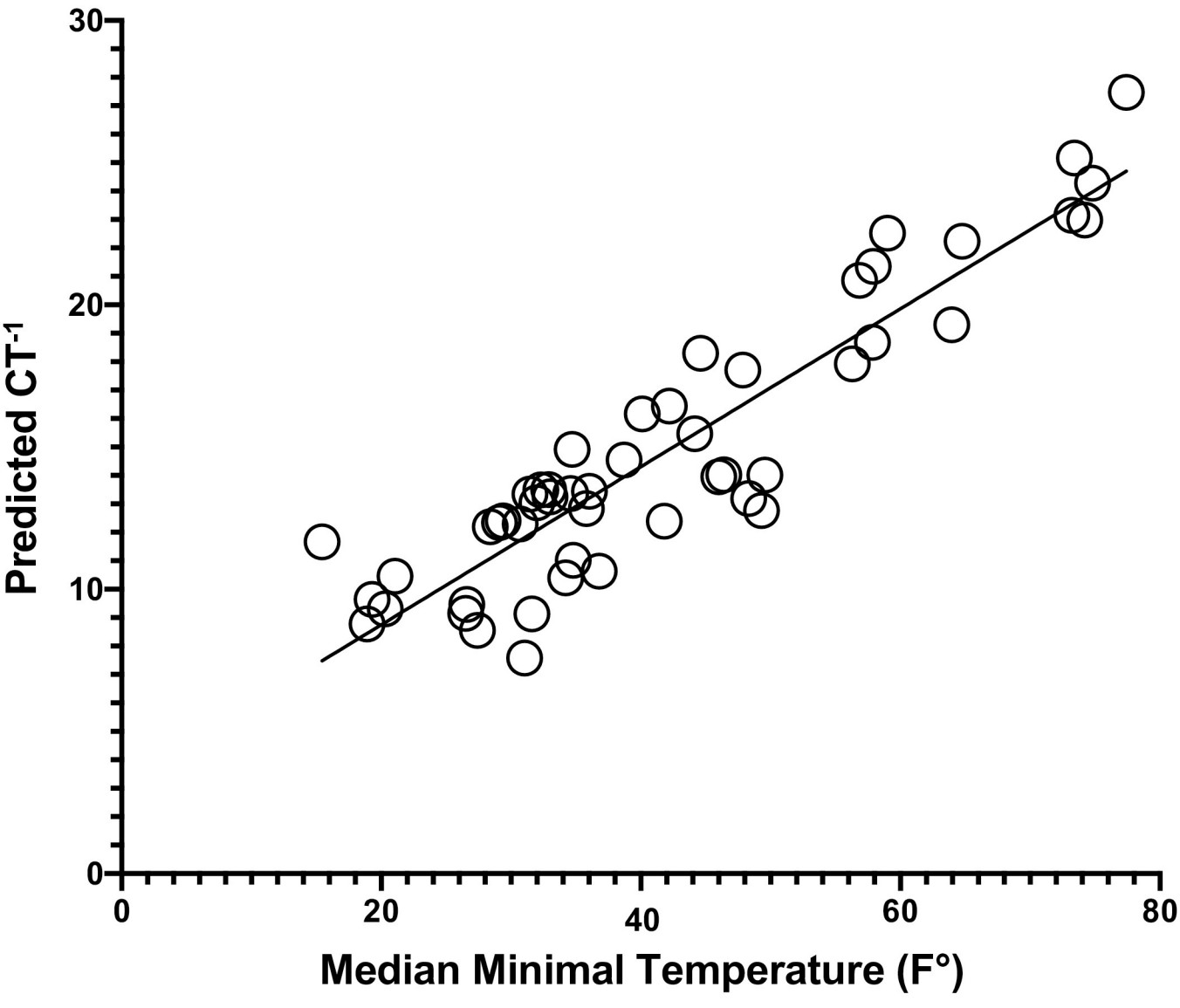

**Fig 2. Shows for fifty representative countries the predicted regression of the reciprocal of the rate of increase of the log of confirmed cases (CT$^{-1}$) vs the median minimal temperature.** Predicted values (i.e. fitted values) are the values that the model predicts using the regression equation generated from the multiple regression. This establishes the relationship between temperature and the rate of daily increase of confirmed cases of COVID-19. The multiple regression of CT$^{-1}$ for all countries yielded a significant correlation (adjusted $R^2$ = 0.610, p = 1.45 x 10$^{-6}$) using covariates Tmin_Med, median age, land area per capita and days of cases.

(from either end of the time period) resulted in only modest changes in the Tmin Med P values, $R^2$, or regression coefficients. Comparing changes from the original analysis using 1/22/20-4/6/20 to results using 1/22/20-3/29/20, 1/22/20-3/21/20, 2/7/20-4/6/20, and 2/24/20-4/6/20 yielded the following findings: 1) P values changed from 2.90x10$^{-6}$ to an average of 4.18x10$^{-6}$ (+/-4.62x10$^{-6}$ SD), 2) $R^2$ changed from 0.60 to an average of 0.63 (+/-0.086 SD), and 3) regression coefficients changed from 0.24 to 0.27 (+/-0.028 SD). Thus, as shown in the S1 and S2 Figs and these summary scores, it is evident that even with 11%-22% changes in either the earlier or later end of the sampled time period, there was very little change in the results using our two step model. This demonstrates the robustness of our findings.

**Table 2. Second regression: The association of temperature and dewpoint with the rate of rise of COVID-19 cases.**

| Case Type | Dependent Variable | Independent Variables | Regression Coefficients | P value | Adjusted $R^2$ |
|---|---|---|---|---|---|
| Confirmed | CT | Tmin | -0.000911 | *** | 0.300 |
| Confirmed | $CT^{-1}$ | Tmin | 0.275 | ******* | 0.449 |
| Confirmed | $CT^{-1}$ | Tmin | 0.205 | **** | 0.587 |
|  |  | Days Cases (DC) | 0.174 | *** |  |
| Confirmed | $CT^{-1}$ | Tmin | 0.247 | ***** | 0.610 |
|  |  | Days Cases (DC) | 0.150 | ** |  |
|  |  | Land Area Per Capita (LAPC) | 21.244 | NS |  |
|  |  | MA | 0.151 | NS |  |
| Case Type | Dependent Variable | Independent Variables | Regression Coefficients | P value | Adjusted $R^2$ |
| Deaths | CT | Tmin | -0.001 | * | 0.172 |
| Deaths | $CT^{-1}$ | Tmin | 0.173 | ** | 0.198 |
| Deaths | $CT^{-1}$ | Tmin | 0.161 | ** | 0.354 |
|  |  | Days Cases (DC) | 0.199 | ** |  |
| Deaths | $CT^{-1}$ | Tmin | 0.164 | ** | 0.323 |
|  |  | Days Cases (DC) | 0.198 | * |  |
|  |  | Land Area Per Capita (LAPC) | -1.058 | NS |  |
|  |  | MA | 0.021 | NS |  |
| Case Type | Dependent Variable | Independent Variables | Regression Coefficients | P value | Adjusted $R^2$ |
| Recovered | CT | Tmin | -0.00105 | *** | 0.303 |
| Recovered | $CT^{-1}$ | Tmin | 0.319 | **** | 0.332 |
| Recovered | $CT^{-1}$ | Tmin | 0.215 | * | 0.442 |
|  |  | Days Cases (DC) | 0.215 | * |  |
| Recovered | $CT^{-1}$ | Tmin | 0.218 | * | 0.421 |
|  |  | Days Cases (DC) | 0.222 | * |  |
|  |  | Land Area Per Capita (LAPC) | 13.446 | NS |  |
|  |  | MA | -0.0117 | NS |  |

The regression coefficients (abbreviated CT for coefficients of time) from the linear regression of the log of cumulative COVID-19 cases vs time (see Table 1) for each country were converted to their reciprocal values ($CT^{-1}$). This was done because of an approximate proportionality of fitted CT with the standardized residuals when performing linear regression with Tmin MEDIAN [31]. A second regression was then performed using the $CT^{-1}$ for each country as the dependent variable and independent variables consisting of median minimal temperature [Tmin_MEDIAN] with or without regional demographic variables (i.e. land area and population median age). The total number of days for each country during which they had cases of COVID-19 between 1/22/20 and 4/6/20 (Days of Cases) was also used as a covariate. Shown are the resulting estimated regression coefficients and the $R^2$ for each regression. To control for multiple comparisons, since the same $CT^{-1}$ and Tmin was used for Confirmed, Deaths and Recovered cases, P values were multiplied by 3. P values are displayed with progressively more stars for each 10-fold decrease in value according to this pattern: 0.05, 0.005, 0.0005, 0.00005, etc = *, **, ***, ****, etc. NS stands for Not Significant (P>0.05). To control for multiple comparisons—since the same $CT^{-1}$ and Tmin was used for Confirmed, Deaths and Recovered cases—p values were multiplied by 3 to account for the repeated measures of cases (See S1 and S2 Tables for further details). All countries were included in the Confirmed Case Analysis. For Death Cases there were 4 countries not included for the following three reasons: 1) Insufficient data for Kuwait (3 days of 0 log cases) and Vietnam (no days of deaths), 2) Inconsistent and limited data for Slovakia (4 days 0 log cases, 10 days no reported cases, then 5 days 0 log cases followed by a single 0.3 log cases), 3) Outlier: Iceland. Recovered Cases had one exclusion: Outlier: Slovakia. Outliers were determined using the ROUT method of Graph Pad's Prism software. Briefly, ROUT first fits a model to the data using a robust method where outliers have little impact. Then it uses a new outlier detection method, based on the false discovery rate, to decide which points are far enough from the prediction of the model to be called outliers [26].

## Analysis of dewpoint and its influence on COVID-19 rate of rise of cases

In addition to regression of $CT^{-1}$ with Tmin_Med, we also examined the median daily DP (DP_Med) for each country. We found that the rate of increase of the log of confirmed cases per unit time using covariates DP_Med, LAPC, MA and Days of Cases, when compared to

Tmin_Med, showed a comparably robust and statistically significant association (adjusted $R^2 = 0.599$, p = 2.68 x10$^{-6}$). Because of the collinear relationship between Tmin_Med and DP_Med, which in linear regression resulted in an adjusted $R^2$ of 0.76, (p = 1.4 x 10$^{-16}$), we assumed that, since Tmin_Med had the higher correlation with cases, it was likely the primary factor underlying the association between weather and COVID-19 transmission.

## Magnitude of effect of temperature change on rate of transmission of COVID-19

From the second regression described above, the following equation was derived:

$$CT = 0.116 - 0.000911 \times Tmin\_Median$$

It was therefore possible to obtain CT for temperatures ranging from 30˚F to 100˚F—i.e. 0.0885 (log cases/day) to 0.0248 (log cases/day), respectively. This analysis indicated that an increase in temperature from 30 to 100˚F was associated with a 72% decrease in the rate of number of confirmed cases per day (CT 0.0885 to 0.0248), whereas a decrease in temperature from 100˚ to 30˚ F represents a 257% increase in the rate of change of confirmed cases (CT 0.0248 to 0.0885). This corresponds to 1.03% decrease in confirmed cases for each degree increase in temperature, and a 3.68% increase for each degree decrease in temperature (e.g. -72/(100–30) and 368/(100–30)).

## Correlation with rates of death and recovered cases of COVID-19

The correlation between the reciprocal of the rise in the log of the number of deaths and recovered cases of COVID-19 ($CT^{-1}$) vs Tmin_Med was 47% and 45% less than that of confirmed cases, respectively (see Table 2). Moreover, the covariates of LAPC and MA for death and recovered cases had regression p values of 0.5 or greater, suggesting that the impact of these covariates on confirmed cases was specific.

## Relationship of CT to doubling time (Td)

As described in the method section, we explored the relationship between CT and doubling time ($T_d$), a standard measure of rate of viral transmission [29]. Doubling times are the intervals at which the cumulative incidence of cases doubles. Under conditions of exponential growth, with a constant growth rate r, the doubling time equals (ln 2)/r [30]. Thus, an increase in the doubling time represents a decreased rate of transmission. We calculated $T_d$ from our data obtained in our first regression, and we took the following steps: we used our first regression of log of confirmed cases vs time to interpolate on which days the number of cases doubled for each country utilizing a centered second order polynomial model to fit the data (mean $R^2 = 0.958$, Standard Deviation = +/- 0.025). Because we were using log of confirmed cases we converted doubling of cases to log form (e.g. 2, 4, 8, 16, 32, 64 corresponds to 0.301, 0.602, 0.903, 1.204, 1.505, 1.806). We included doubling times up to day 64. We calculated the average $T_d$ (i.e. number of days) between each interval corresponding to a doubling of confirmed cases. We then performed a linear regression on $T_d$ and CT, and $T_d$ and Tmin Med, with the following results:

$$T_d = 7.56 - 49.9 * CT (\text{adjusted } R^2 = 0.724, \ p = 3.2x10^{-15})$$

$$T_d = 1.01 + 0.064 * Tmin\_Med (\text{adjusted } R^2 = 0.45, \ p = 1x10^{-7})$$

For all 50 countries the average $T_d$ for the first 6 doubling times was 3.69 (range 1.28–8.65 days).

Thus, the smaller the CT (i.e. rate of increase of cases of COVDI-19) or greater the Tmin Med, the longer the doubling time—consistent with increasing temperatures decreasing the rate of transmission.

## Details of the model formulations (including equations in appropriate context)

We began by performing a two-step regression to derive the relationship between temperature and rate of increase in COVID-19 cases.

First regression involved the rate of rise of confirmed cases of COVID-19 in 50 countries, and the associated local meteorological and demographic variables:

$$\text{Log(Cumulative Cases)} = CT * Day - 3416$$

A second regression was then performed using the inverse of CT for each country as the dependent variable and the independent variables consisted of associated meteorological variables (i.e. median minimal temperature [Tmin_Med] or median dewpoint [DP]) with or without regional demographic variables (i.e. land area per capita and population median age):

$$CT^{-1} = 0.278 * \text{Tmin\_Med} + 3.19$$

## Discussion

The major finding of this study is that the rates of transmission of SARS-CoV-2 infections are robustly associated with the ambient atmospheric temperature. The results of our analysis of longitudinal data from 50 representative countries in the Northern Hemisphere suggest that changes in the minimum daily temperature are associated with alterations in SARS CoV-2 transmissibility and viability. In addition, we found that the median daily dewpoint had a significant effect on the cumulative confirmed COVID-19 cases over time. However, because of the collinearity between dewpoint and atmospheric temperature over time, it was not possible to deduce whether the two atmospheric parameters independently affect SAR CoV-2 transmission. Since dewpoint is less predictable, and temperature more frequently monitored and used as a meteorological variable, we focused on changes in temperature for its potential predictive value.

We found that in 50 representative countries spread out over 70% of the possible latitude and longitude across the Northern Hemisphere, a slower rise in the log of daily cumulative number of confirmed cases was associated with higher daily minimum temperatures and dewpoints over the period from 1/22/20 to 4/6/20. This finding was robust, yielding an adjusted $R^2$ = 0.610, p = 1.45 x $10^{-6}$ (using covariates Tmin_Med, median age, land area per capita, and days of cases). The correlation between the reciprocal of the rise in the log of the daily number of death and recovered cases ($CT^{-1}$) vs Tmin_Med was almost 50% less than that of confirmed cases. This is not surprising, given that the impact of temperature on seasonal respiratory viruses has been found to have a consistent influence on the stability and transmission of infection, and not on the morbidity of infected individuals [2]. In view of these findings, atmospheric temperature would be expected to influence confirmed cases, and not deaths or the number of recovered subjects (e.g. once an infection occurred, temperature exerts a small or absent effect on the course of COVID-19 morbidity).

This study was able to achieve a clear and robust association between meteorological changes (i.e. temperature and dewpoint) and transmissibility of COVD-19, and to quantitate the magnitude of these correlations, because of the following steps taken: 1. we chose systematic criteria to select 50 representative countries that were selected to increase the probability that the data gathered (e.g. confirmed, deaths, recovered) and the rate of spread of COVID-19 would be comparable or at least randomly biased; 2. we took great care to ensure that we used weather stations located as close to the longitude and latitude corresponding to the cases in each country as presented by the Johns Hopkins real time COVID-19 tracking database; 3. our two-step regression model permitted us to relate the rate of rise of cases to changes in meteorological variables; 4. we focusing our analysis on the initial rise in cases across the various countries. During this early phase there was a delayed response on the part of the country (usually both the government and the populace) to institute and practice non-pharmaceutical interventions to slow the spread of the virus. We were therefore able to capture a period where weather and not social interventions played the predominant role in impacting transmission. And 5. after investigating numerous independent variables, we were able to select those that mediated the greatest association between meteorological variables and rate of transmission in our multivariable regression model.

Although our study is a systematic and quantitative analysis of the dependence of SAR-CoV-2 infection rates on temperature dewpoint that has demonstrated a robust and significant correlation (Table 2), it has several notable limitations. This is an ecological study and is therefore potentially subject to ecological fallacy [35]. Moreover, our data are not direct measures of individual-to-individual transmission, so cause and effect relationships cannot be established. However, the biological plausibility for our hypothesis is supported by previous work showing that the transmission rates of SARS-CoV-2 and similar viruses depend upon atmospheric temperature [2, 36].

In temperate regions, the annual occurrence of respiratory viral diseases during the winter season—from the common cold to influenza—has been appreciated for several thousand years [2]. A similar seasonal pattern of infections has been reported for SARS-CoV, which was prevalent mostly during winter months [2]. Despite this, there is sparse evidence on the seasonal behavior of the novel SARS-CoV-2 [6], and there are conflicting reports on how its transmission is affected by meteorological conditions. In this context, our study provides the most comprehensive and up-to-date evidence for a robust and significant impact of temperature and dewpoint on SARS-CoV-2 transmissibility.

Further research is needed to clarify whether the association between temperature and SARS-CoV-2 transmissibility described here has a biological underpinning, and whether our model is accurate for seasonal predictions, surveillance and preparedness related to the spread of the virus over different geographical areas. Nonetheless, should the association between temperature and the rate of transmission hold true (either because of an underlying causal relationship or because of the empirical relationship described here), it may be possible to approximate the impact of weather on the rates of transmission of SARS-CoV-2. Because atmospheric parameters are measurable, and routinely tracked on a daily basis across the world, determining the contribution of temperature to the rate of transmission should make it possible to quantify the contribution of non-weather factors (e.g. social and non-pharmaceutical interventions (NPI))—since observed rate of transmission is equal to the contribution of meteorological variables plus the contribution of NPI. Once calibrated to the contributions of meteorological variables and social interventions and other NPI, such estimates may be able to inform and guide systematic academic, business and governmental decisions about when and how to impose or relax shelter-in-place or social distancing guidelines. Such evidence-based policies may succeed in minimizing social and economic disruptions due to the pandemic

while at the same time optimizing public health and safety. Finally, our analysis predicts that, between the range of 30 to 100˚F, the decrease in the rate of COVID-19 transmission with increasing temperature (1% per degree F) is smaller than the increase rate of transmission with decreasing temperatures (3.7% per degree F) in the 50 representative countries in the Northern Hemisphere countries examined. If this model is correct, then it implies that effort invested in containing, minimizing, and ideally eliminating the spread of COVID-19 during Spring and Summer months could pay off significantly in the Fall and Winter, due to potentially disproportionate effects of decreasing temperatures compared to increasing temperatures on the current pandemic.

## Supporting information

**S1 Table. The regression coefficients (abbreviated CT for coefficients of time) from the linear regression of the log of cumulative COVID-19 cases vs time (see Table 1) for each country were converted to their reciprocal values (CT⁻¹).**
(DOCX)

**S2 Table. Post-hoc analysis of multivariable regression for temperature.**
(DOCX)

**S3 Table. Post-hoc analysis of multiple regression for dewpoint.**
(DOCX)

**S1 Fig.**
(TIF)

**S2 Fig.**
(TIF)

## Acknowledgments

The authors would like to acknowledge the intellectually stimulating discussions and consultations with Howard Burkom and Steven Babin, and the statistical consultation and mentorship provided by Howard Burkom.

## Author Contributions

**Conceptualization:** Adam Kaplin, Caesar Junker, Mary Anne Ribeiro, Aruni Bhatnagar.

**Data curation:** Eileen Yu, Ted Smith, Shesh N. Rai.

**Formal analysis:** Adam Kaplin.

**Investigation:** Adam Kaplin, Caesar Junker, Ted Smith.

**Methodology:** Adam Kaplin, Mary Anne Ribeiro, Shesh N. Rai.

**Project administration:** Adam Kaplin, Anupama Kumar, Ted Smith.

**Software:** Anupama Kumar.

**Supervision:** Adam Kaplin, Aruni Bhatnagar.

**Validation:** Adam Kaplin, Michael Wang, Shesh N. Rai.

**Visualization:** Adam Kaplin.

**Writing – original draft:** Adam Kaplin.

**Writing – review & editing:** Caesar Junker, Anupama Kumar, Mary Anne Ribeiro, Eileen Yu, Michael Wang, Ted Smith, Shesh N. Rai, Aruni Bhatnagar.

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
