## [Decision Letter · Decision Letter 0]

18 Aug 2020

PONE-D-20-17772

Evidence and magnitude of seasonality in SARS-CoV2 transmission: Penny wise, pandemic foolish?

×

PLOS ONE

Dear Dr. Kaplin,

Thank you for submitting your manuscript to PLOS ONE. After careful consideration, we feel that it has merit but does not fully meet PLOS ONE’s publication criteria as it currently stands. Therefore, we invite you to submit a revised version of the manuscript that addresses the points raised during the review process.

The Authors are expected to address all the criticisms by all Reviewers. In particular, avoid implication on causality, justify, or avoid exclusion of countries, including countries in the Southern Hemisphere (Reviewers #2 and #3), improve the writing of the manuscript, revise Figure 2 for accuracy and clarify (Reviewer #1), assess the impact of the choice of time period on the results (Reviewer #2), avoid conclusion on seasonality, and provide the equations for the major models used for the analysis (Reviewer #3). In additional to the above comments, please address,

L87-90, the exclusion of China, Russia and Italy is not well justified. The case numbers in different countries are available from different sources which were translated into English. There are other exclusions which were not fully justified.Please remove citations 6 and 7Carry out relevant sensitivity analysis to justify the conclusion that the results were robust.

We look forward to receiving your revised manuscript.

Kind regards,

Eric HY Lau, Ph.D.

Academic Editor

PLOS ONE

Journal Requirements:

2. Please refer to the specific statistical analyses performed as well as any post-hoc corrections to correct for multiple comparisons. If these were not performed please justify the reasons. Please refer to our statistical reporting guidelines for assistance (https://journals.plos.org/plosone/s/submission-guidelines.#loc-statistical-reporting), and please ensure that the statistical analysis is fully reported in the Methods section.

3. Please consider modifying your title to ensure that it is specific, descriptive, concise, and comprehensible to readers outside the field.

4.We note that you have indicated that data from this study are available upon request. PLOS only allows data to be available upon request if there are legal or ethical restrictions on sharing data publicly. For information on unacceptable data access restrictions, please see http://journals.plos.org/plosone/s/data-availability#loc-unacceptable-data-access-restrictions.

5.Thank you for stating the following in the Competing Interests section:

[The authors have declared that no competing interests exist.].   

We note that one or more of the authors are employed by a commercial company: PUPA

Additional Editor Comments (if provided):

The Authors are expected to address all the criticisms by all Reviewers. In particular, avoid implication on causality, justify, or avoid exclusion of countries, including countries in the Southern Hemisphere (Reviewers #2 and #3), improve the writing of the manuscript, revise Figure 2 for accuracy and clarify (Reviewer #1), assess the impact of the choice of time period on the results (Reviewer #2), avoid conclusion on seasonality, and provide the equations for the major models used for the analysis (Reviewer #3). In additional to the above comments, please address,

1. L87-90, the exclusion of China, Russia and Italy is not well justified. The case numbers in different countries are available from different sources which were translated into English. There are other exclusions which were not fully justified.

2. Please remove citations 6 and 7

3. Carry out relevant sensitivity analysis to justify the conclusion that the results were robust.

Reviewers' comments:

Reviewer's Responses to Questions

**Comments to the Author**

1. Is the manuscript technically sound, and do the data support the conclusions?

Reviewer #1: No

Reviewer #2: Partly

Reviewer #3: Partly

2. Has the statistical analysis been performed appropriately and rigorously? 

Reviewer #1: No

Reviewer #2: No

Reviewer #3: Yes

3. Have the authors made all data underlying the findings in their manuscript fully available?

Reviewer #1: No

Reviewer #2: Yes

Reviewer #3: Yes

4. Is the manuscript presented in an intelligible fashion and written in standard English?

Reviewer #1: No

Reviewer #2: Yes

Reviewer #3: Yes

5. Review Comments to the Author

Reviewer #1: If only it were that simple! On the whole, linear regressions can generate hypotheses, but they can't test them in the way that you are attempting to do. Figure 2 looks very impressive until you realize that it is extracted from a model where you've got quite a few variables, and not that many data points. The correlation shown is - or seems to be - meaningful and important. The raw and derived data (including your CT measure) is much less impressive.

I may have misunderstood Figure 2, but then it is very poorly explained – see my comments below.

I hesitate to criticize you for this, because a lot of well-known epidemiologists are guilty of the same thing, but it would help a lot if you were clearer about the phenomena that you are modelling, and what we don’t know. Epidemiologists use very simple models where they assume that someone who is infected always becomes symptomatic after a certain interval. This is in reality not the case. A very interesting series of articles were published recently by Jeff Shaman’s group at Columbia University who sampled the population of New York City, looking for the presence of respiratory viruses. They found that the majority of people carrying detectable virus had no symptoms. They also found that as many people had detectable virus in summer as in winter. I feel your article (or similar articles in future) would be improved if you thought more about the possible mechanisms that give rise to seasonality, and were clearer about what we DON’T know. I’ve put some references below.

Can I ask that in future you take a little more trouble in writing your articles? There are lots of problems with this manuscript. For a start, there are many sentences that don't have verbs in them. For example, p8, line 119. In the abstract, P2, lines 44-45 don't make sense. Then on the most important item in your paper, Figure 2, the vertical axis is not labelled correctly. In fact Fig 2 says that as temp increases the number of cases increases! You are also inconsistent in abbreviations - sometimes you write DP, other times DEWP. It's not fair on the reviewers to ask them to put in a lot of effort if you don't make your article easy to read.

There is also a lot of repetition in the article between the table and figure legends and the main text. Just write everything once, clearly. I always start by writing the legends, because they are what most people read first. Either put the detail in there, or refer, in the legends, to the main text for the detail. Most people write the main text first, then they realize that they need to explain the figures, and write the same material over again - I think this might be what you have done.

Galanti, M., et al. "Rates of asymptomatic respiratory virus infection across age groups." Epidemiology & Infection 147 (2019).

Galanti, M, et al. "Longitudinal active sampling for respiratory viral infections across age groups." Influenza and Other Respiratory Viruses 13.3 (2019): 226-232.

Shaw Stewart, PD. Seasonality and selective trends in viral acute respiratory tract infections. Medical Hypotheses 2016; 86 104–119. https://www.douglas.co.uk/f_ftp1/ShawStewart_final_1-s2.pdf

Reviewer #2: The paper is of interest and I would like to see a number of improvements to this work before it can be published.

The paper looks at the association between the COVID-19 case number and the temperature. This topic is not novel and has been widely studied (e.g., [1-5]). However, the current manuscript does not give a clear and comprehensive review of the most up-to-date background and studies of this topic. The author should clearly address the innovative contribution of their study and compare with other relevant studies (e.g., whether a new method is used? what is the advantage of this new method? Does the study cover a more comprehensive dataset? Does the study give any new conclusion or verify existing results?) These should be discussed in the Introduction and Discussion of this manuscript.

In the Method section, the authors state that they exclude data from China and Russia and list the relevant news report as a support. It is generally acceptable to cite information from the media for the COVID study when the information is simply an objective report of what has happened. However, the citation 6 and 7 in the manuscript are based on some so called “source of the leak” and cannot be verified. It is just a speculation made by someone not from the scientific community. The WHO has never made a similar statement. Therefore, it is not scientific to exclude data from China and Russia based on current “support”. The authors should cite reliable and objective information (e.g., published papers, as the author already do to explain Italy) or give some more convincible reasons. Similarly, I am interested to know the reason of excluding Canada and the United States of America.

The authors state that the method used in this study is robust. By looking at the Figure 1, I doubt if the conclusion made in the manuscript is really robust. The cumulative number of COVID 19 cases generally shows a concave trend. Hence, if you look at the first half of the time period, the coefficients of time (CT) will be larger. Similarly, the CT will be smaller if you look at the other half of the time period. It is therefore needed to investigate whether the conclusion still holds when the time period changes. If the conclusion is largely affected, this point should be stated as a caution in the manuscript. Meanwhile, the author may consider including the quadratic term in the first regression.

The authors should be cautious when saying “a one degree increase leads to a 1% decrease--and a one degree decrease leads to a 3.7% increase” in the Abstract. “Lead” normally indicates causality. It should be replaced by “associate”.

Minor Comments:

In line 57, it should be “Coronavirus” rather than “Corona VIrus”

In Figure 2, the y axis label should be “reciprocal of CT (CT-1)” rather than “confirmed cases (Log)”.

Additional Reference:

1. Wu Y, Jing W, Liu J, et al. Effects of temperature and humidity on the daily new cases and new deaths of COVID-19 in 166 countries. Sci Total Environ. 2020;729:139051. doi:10.1016/j.scitotenv.2020.139051

2. Shi P, Dong Y, Yan H, et al. Impact of temperature on the dynamics of the COVID-19 outbreak in China. Sci Total Environ. 2020;728:138890. doi:10.1016/j.scitotenv.2020.138890

3. Xie J, Zhu Y. Association between ambient temperature and COVID-19 infection in 122 cities from China. Sci Total Environ. 2020;724:138201. doi:10.1016/j.scitotenv.2020.138201

4. Ujiie M, Tsuzuki S, Ohmagari N. Effect of temperature on the infectivity of COVID-19. Int J Infect Dis. 2020;95:301-303. doi:10.1016/j.ijid.2020.04.068

5. Sajadi MM, Habibzadeh P, Vintzileos A, Shokouhi S, Miralles-Wilhelm F, Amoroso A. Temperature, Humidity, and Latitude Analysis to Estimate Potential Spread and Seasonality of Coronavirus Disease 2019 (COVID-19). JAMA Netw Open. 2020;3(6):e2011834. Published 2020 Jun 1. doi:10.1001/jamanetworkopen.2020.11834

Reviewer #3: Thank you for giving me an opportunity to review this submission titled “Evidence and magnitude of seasonality in SARS-CoV2 transmission: Penny wise, pandemic foolish?”. The study is very timely given the current pandemic situation and indeed tackles an important issue, and to some extent controversy, about environmental and atmospheric conditions conducive for COVID-19 transmission. I acknowledge that the article has its own merits and immense potential to be able to contribute to current body of knowledge; however, there are still quite a few issues I found perplexing and needs to be addressed better before I give my full recommendation for publication. The first thing that stands out is the use of the work “seasonality” which to me might be a little bit misleading as effectively only one season was evaluated. It is a strong claim to make given very limited evidence presented in the study and it did not help that the design further limited this viewpoint by not considering Southern Hemisphere countries/regions. Rather, a more appropriate wording must have focused around correlation between meteorological or environmental or atmospheric conditions with COVID-19 transmission. I have some more specific questions and concerns detailed below. Pending satisfactory responses, I can then give my recommendation to have this published.

Abstract:

• Line 38: Can you expound more, either in the Methods or Results section, how the reciprocal of CT can be associated with doubling time? If possible, is there some exact mathematics as to how they are related?

• Lines 49-50: While it does make sense to make the claim about summer months, I find it hard to make a claim on winter months given there is no actual evidence to support this. Data were limited to the Northern Hemisphere countries which are yet to experience winter, and all that we have are claims based on temperature changes which is not fully indicative of a seasonal change. Actually, this applies to the summer claim as well. Perhaps, this statement can be restated so that it does not appear to be overpromising and just stick with correlations between temperature or humidity with transmission?

Methods:

• Lines 84-87: However, my take is that wouldn't using Southern Hemisphere countries even as controls be also helpful, as at the same time they experienced drop in temperatures? Especially, if the study is also making a claim about possible 'resurgence' during colder months? In fact, I am not convinced why does the data cannot from two hemispheres cannot be pooled.

• Lines 87-90: While not so much of an issue at the moment, but have you considered exploring or conducting some kind of sensitivity analyses to check how the results would have changed if these countries are added in the analyses – at least even just with Italy?

• Lines 91-92: What did you mean by countries with separate reports of cases from multiple regions? Can you list what these countries are?

• Line 93: How did you assess reliability of COVID-19 and meteorological data?

• Lines 107-111: Were there opportunities to also used other variables? While it does help that you provided a reference as to why Tmin is one of the most appropriate to use in this context, I would be inclined to also explore other measures (such as Tmean, Tmed, Tmax) and see how the correlation magnitudes change or not. To my mind, this could have presented a more comprehensive picture of how temperature is correlated with transmission. This also applies to other atmospheric data which NOAA is tracking and ‘complete’ for all countries anaysed.

• Lines 113-114: I’m sorry but I did not understand this fully. So does this mean whenever there were 0 cases in a day, corresponding Tmin and DP were not also recorded? Why is this so? Even if, say, there is one day that no confirmed cases were reported squeezed between two days with high cases, that day with 0 case is already dropped? But I still don’t see why this is needed to be performed given it is the reciprocal of CT (based on cumulative counts) that is being modelled. So in other words, the entire span of the series for CT per country should also be the same timeline of the temperature and dewpoint data to be considered. Or maybe this is already performed, but I am just understanding this incorrectly? Can you please explain further? Best if you can give me an actual scenario/example?

• Can you add a subsection describing with thorough details the model formulations (including even just ‘generic’ equations but appropriately contextualised)?

Results:

• Lines 150-152: Are there other measures you can consider to assess the goodness-of-fit? To my mind, while R2 is okay, we will actually expect it to be very high given we are fitting to a function of cumulative confirmed cases which are naturally increasing in in time. So no matter what the shape is, especially with log(cumulative cases), the linear increasing trend will be captured very well. I will be more inclined to report other goodness-of-fit measures too just as an added support to this great fit.

• Figure 2: It seems to me that there is a systematic bias here, where for higher values the best-fit (predicted) values are always less than those? In other words, for higher confirmed inverse CT there appears to be a systematic underestimation. Wouldn’t this affect model interpretation and appropriateness for higher values of either inverse CT or temperature?

• Lines 224-225: How often do you see this much change in temperature (drop or increase of 70°F)? To me, I would prefer to present them in a much more interpretable format, say the relative size of temperature changes going from one latitudinal zone to another. Or even simpler, in around 5-10°F changes?

Discussion:

• Line 240: How do you qualify ‘strong and robust’ with your findings? To me, I feel that this is such a strong claim and a bit of an overstretch given the issues I pointed out earlier about data catchment and quality as well as standard correlational approaches via regression. Perhaps, consider rephrasing this so as not to oversell the results?

• Line 242: Again, I find ‘impacts’ a bit too strong given there is no causality established - what we only know is that there is statistical evidence (from correlations) that they are related.

• Line 272: In claiming similarities in seasonal pattern of infections with SARS-CoV, perhaps you can add more context into this. As your study focused only on temperature (and to some extent, humidity) perhaps it is better if you discuss this in the context of changes in temperature (or ranges) here from summer to winter. Again, it is hard to make claims about summer and winter differences when all your data are based on summer and the only proxy for winter conditions is based on temperature. Furthermore, I suppose this seasonal pattern can be different in many regions, and what happens for tropical climates where there is no winter?

• Lines 293-297: Can you further explain this sentence? Of course, they will be different from a calculation perspective given you have different denominators in quantifying changes between increasing-to-decreasing and decreasing-to-increasing even for the same magnitude (70°F) of difference. To my mind, they are all based on the estimated regression coefficients so in principle the changes should be similar.

• Line 299: Again, I find the claim that “could pay off significantly in the fall and winter” a bit of an overstretch given the same issues I raised before about not having analysed winter conditions (e.g., by not considering the Southern Hemisphere).

Some minor comments:

• In general, when presenting temperatures, would you consider mentioning them in °C too, especially for audience who use this scale? This adds more context in the study.

• Also, maybe reconsider restructuring the Methods and Results sections, as some contents are better written under Methods (e.g. model-fitting stages) while others are in Results (e.g. Table 1).

• Line 60 (Introduction): I believe you meant ‘MERS-CoV’ here?

• Lines 65-66 (Introduction): Perhaps you can expound a bit more on these evidence or suggestions, especially its quality? How about counter evidence or arguments?

• Line 79 (Methods): What do longitudinal changes mean? By time? Perhaps you can use a different word to make this clearer.

• Lines 95-97 (Methods): Perhaps clearly mention that you are referring to ‘daily’ (new) confirmed cases.

• Lines 100-102 (Methods): To be honest, I am more inclined to not discuss anything about longitude coverage given it does not affect meteorological conditions, right? This is to save space or number of words.

• Lines 173-175 (Results): I think it would have been more appropriate (but admittedly, more complex) if the counts have been age-standardised with respect to a reference population? So no need for such covariate? Has this been explored/considered?

• Table 2: I believe the 4th column refers to estimated regression coefficients right? Perhaps just mention this instead of correlation as it might be a bit misleading?

• Table 2: I am not too keen about reporting analyses with Recovered data in the same amount of details as with Confirmed Cases and Deaths just because they may be of the poorest quality among the three, at least in many countries that I know of, Recoveries are almost not always reported. So this might give an impression that the data is also robust if included as part of the main analysis. May I suggest instead to remove this in the Table and just simply write out a paragraph for somewhere in the Results section.

• Table 2: In my experience, annotating p-values > 0.05 is not a standard wat of reporting. Oftentimes you annotate those which are statistically significant, not the other way around. Also, the use of scientific notation/exponentiation in p-values appear to obscure the values p-values (making them appear less in magnitude), so may I suggest changing to 3 or 4 decimal places and whenever less than 0.001 just say <0.001?

• Lines 236-237 (Results): Shouldn’t this be part of Methods?

• Lines 279-281 (Discussion): I think the way it is currently written it does not make sense. Longitudinal differences do not dictate atmospheric and meteorological conditions, so if the reference is to the model using temperature then it does not apply. But if the reference is to the approach but using difference covariates (say, cultural or population behaviour measures) then I see how it can also be useful. What I am trying to say is perhaps you can rephrase this better to avoid this ambiguity in interpretations.

6. PLOS authors have the option to publish the peer review history of their article (what does this mean?). If published, this will include your full peer review and any attached files.

Reviewer #1: No

Reviewer #2: No

Reviewer #3: No

---

## [Author Response · Author response to Decision Letter 0]

28 Sep 2020

*****Response to Reviewers:

• Please find below our rebuttal letter that responds to each point raised by the academic editor and three reviewers. 

• To ensure we systematically addressed all issues raised, we added our responses to the original email we received and highlighted in yellow all responses. 

• In addition to highlighting, we have bracketed our responses between “*****”.

• We believe that our responses to the editors and reviewers comments, and the changes we made accordingly have improved our manuscript considerably, and we appreciate the guidance and critiques. *****

Tuesday, August 18, 2020 at 12:56:20 Eastern Daylight Time

Subject: PLOS ONE Decision: Revision required [PONE-D-20-17772ti - [EMID:7c5f4de3c97861cfti

Date: Tuesday, August 18, 2020 at 5:04:40 AM Eastern Daylight Time

From: em.pone.0.6d6037.df18d559@editorialmanager.com on behalf of PLOS ONE

To: Adam Kaplin

PONE-D-20-17772

Evidence and magnitude of seasonality in SARS-CoV2 transmission: Penny wise, pandemic foolish?

PLOS ONE

Dear Dr. Kaplin,

Thank you for submitting your manuscript to PLOS ONE. After careful consideration, we feel that it has merit but does not fully meet PLOS ONE’s publication criteria as it currently stands. Therefore, we invite you to submit a revised version of the manuscript that addresses the points raised during the review process.

*****We greatly appreciate kind words about the merit of our manuscript as well as the opportunity to address all the editors and reviewers comments--we believe in doing so, we have been guided towards a much better submission that meets PLOS ONE’s publication criteria*****

The Authors are expected to address all the criticisms by all Reviewers. In particular, avoid implication on causality, justify, or avoid exclusion of countries, including countries in the Southern Hemisphere (Reviewers #2 and #3), improve the writing of the manuscript, revise Figure 2 for accuracy and clarify (Reviewer #1), assess the impact of the choice of time period on the results (Reviewer #2), avoid conclusion on seasonality, and provide the equations for the major models used for the analysis (Reviewer #3). In additional to the above comments, please address,

*****We have removed implications of causality, justified the exclusion and inclusion criteria for countries (including the Southern Hemisphere), improved the writing of the manuscript by addressing the 3 errors found by Reviewer 1, revised Figure 2, provided an explanation for the time period chosen, performed a sensitivity analysis to assess the impact of the choice of time period and the robustness of our findings, addressed the issue of seasonality, provided s subsection that included the equations for the major models used in our analysis, and all other reviewer comments that were raised. *****

1. L87-90, the exclusion of China, Russia and Italy is not well justiﬁed. The case numbers in diﬀerent countries are available from diﬀerent sources which were translated into English. There are other exclusions which were not fully justiﬁed.

*****We provided justification for all countries included and the ones that were excluded and mentioned by the editors and the reviewers. *****

2. Please remove citations 6 and 7

*****Citations 6&7 were removed and replaced by references from the literature. *****

3. Carry out relevant sensitivity analysis to justify the conclusion that the results were robust.

*****We performed a sensitivity analysis, included the discussion of a summary of the results in the manuscript and included two figures in the supplementary material section demonstrating the actual individual results from the four additional analyses we did by changing the dates from which cases and temperatures were drawn. The end result was that the sensitivity analysis demonstrated no meaningful changes despite 22% change in the days on which cases and temperatures were recalculated. Thus, we justified the conclusion that the results were robust. *****

Please include the following items when submittng your revised manuscript:

A rebuttal letter that responds to each point raised by the academic editor and reviewer(s). 

You should upload this letter as a separate ﬁle labeled 'Response to Reviewers'.

A marked-up copy of your manuscript that highlights changes made to the original version. You should upload this as a separate ﬁle labeled 'Revised Manuscript with Track Changes'.

An unmarked version of your revised paper without tracked changes. You should upload this as a separate ﬁle labeled 'Manuscript'.

If you would like to make changes to your ﬁnancial disclosure, please include your updated statement in your cover letter. Guidelines for resubmittng your ﬁgure ﬁles are available below the reviewer comments at the end of this letter

If applicable, we recommend that you deposit your laboratory protocols in protocols.io to enhance the reproducibility of your results. Protocols.io assigns your protocol its own identiﬁer (DOI) so that it can be cited independently in the future. For instructions see: hkp://journals.plos.org/plosone/s/submission-guidelines#loc- laboratory-protocols

We look forward to receiving your revised manuscript. Kind regards,

Eric HY Lau, Ph.D. Academic Editor PLOS ONE

Journal Requirements:

1. Please ensure that your manuscript meets PLOS ONE's style requirements, including those for ﬁle naming. The PLOS ONE style templates can be found at

hkps://journals.plos.org/plosone/s/ﬁle?id=wjVg/PLOSOne_forma[ng_sample_main_body.pdf and hkps://journals.plos.org/plosone/s/ﬁle?id=ba62/PLOSOne_forma[ng_sample_title_authors_aﬃliations.pdf

2. Please refer to the speciﬁc statistical analyses performed as well as any post-hoc corrections to correct for multiple comparisons. If these were not performed please justify the reasons. Please refer to our statistical reporting guidelines for assistance (hkps://journals.plos.org/plosone/s/submission-guidelines.#loc-statistical-reporting), and please ensure that the statistical analysis is fully reported in the Methods section.

*****The following was performed to account for multiple comparisons, and included in the text of the manuscript. The unit variable was country. At the until level, dependent and independent variables were evaluated. There were three correlated outcome measures: cases that were confirmed, deaths or recovered. For each of these three outcomes, an aggregate measure CT/CT-1 was calculated and then associated with multiple independent variables. Due to evaluating three outcomes in the same population, a multiple adjustment was required. A simple and conservative approach has been used to compare observed p values (p) with adjusted alpha (=alpha/3) or inflate the observed p values by 3 (p*3) and compare at fixed alpha level.(1, 2) We adopted the latter approach. Also note that because CT and CT-1 are one-to-one transformations of one another, there was no need to adjust for using CT or CT-1 as an dependent variable. *****

3. Please consider modifying your title to ensure that it is speciﬁc, descriptive, concise, and comprehensible to readers outside the ﬁeld.

*****The title has been modified as recommended*****

4. We note that you have indicated that data from this study are available upon request. PLOS only allows data to be available upon request if there are legal or ethical restrictions on sharing data publicly. For information on unacceptable data access restrictions, please see hkp://journals.plos.org/plosone/s/data-availability#loc- unacceptable-data-access-restrictions.

*****There is some confusion here. All of the data is publicly available without restrictions. Here is what the authors indicated regarding data: 

“The manuscript describes where all data was obtained, and all data is publicly available. The corresponding author will gladly make the data available to anyone who would like a copy of the data collected from the publicly available websites if it will save time for anyone who requests.” [Emphasis added]

The key points are this: 

1. All data was obtained from websites that are publicly available without restrictions. 

2. In the manuscript we specifically provide all the information needed to get to the online storage site and download all of the data used in the manuscript.

3. All we were saying regarding data being “available upon request” is that if a reader is too busy to go to the website themselves and collect the data, we would happily provide the data already collected from the online site. In other words—the reader can get it themselves, but we were offering to make the task less onerous by providing the data for them. 

Where the journal asks for the authors to “Describe where the data may be found in full sentences.” We have removed #3 above for clarification and made it abundantly clear that all data is available, unrestricted and at the sites we reference in our manuscript. 

We also submitted files that contained all of the data that was used for our analyses. *****

a) If there are ethical or legal restrictions on sharing a de-identiﬁed data set, please explain them in detail (e.g., data

contain potentially identifying or sensitive patient information) and who has imposed them (e.g., an ethics committee). Please also provide contact information for a data access committee, ethics committee, or other institutional body to which data requests may be sent.

b) If there are no restrictions, please upload the minimal anonymized data set necessary to replicate your study ﬁndings as either Supporting Information ﬁles or to a stable, public repository and provide us with the relevant URLs, DOIs, or accession numbers. Please see hkp://www.bmj.com/content/340/bmj.c181.long for guidelines on how to de-identify and prepare clinical data for publication. For a list of acceptable repositories, please see hkp://journals.plos.org/plosone/s/data-availability#loc-recommended-repositories.

We will update your Data Availability statement on your behalf to reﬂect the information you provide.

*****There are no restrictions and we have uploaded the data set necessary to replicate our study as supporting information, in addition to the tables already provided in our original submission that contain all of the data used for the multivariable regression analysis. *****

[The authors have declared that no competing interests exist]

We note that one or more of the authors are employed by a commercial company: PUPA

1. Please provide an amended Funding Statement declaring this commercial aﬃliation, as well as a statement regarding the Role of Funders in your study. If the funding organization did not play a role in the study design, data collection and analysis, decision to publish, or preparation of the manuscript and only provided ﬁnancial support in the form of authors' salaries and/or research materials, please review your statements relating to the author contributions, and ensure you have speciﬁcally and accurately indicated the role(s) that these authors had in your study. You can update author roles in the Author Contributions section of the online submission form.

*****The author who is affiliated with Pupa only provided intellectual input into the design and execution of portions of the study. She did so independently of her affiliation with Pupa. Pupa is a company specializes in early childhood education in Brazil (0-6 years of age), and neither funds nor is remotely involved in anything related to any aspect of this submission. Pupa merely was mentioned as being relevant to the authors affiliation, but the co-author was acting independently of that company. *****

“The funder provided support in the form of salaries for authors [insert relevant initials], but did not have any additional role in the study design, data collection and analysis, decision to publish, or preparation of the manuscript. The speciﬁc roles of these authors are articulated in the ‘author contributions’ section.”

If your commercial aﬃliation did play a role in your study, please state and explain this role within your updated Funding Statement.

2. Please also provide an updated Competing Interests Statement declaring this commercial aﬃliation along with any other relevant declarations relating to employment, consultancy, patents, products in development, or marketed products, etc.

*****There was absolutely no funding for any aspect of this study and there are no competing interests for any of the authors. *****

Within your Competing Interests Statement, please conﬁrm that this commercial aﬃliation does not alter your adherence to all PLOS ONE policies on sharing data and materials by including the following statement: "This does not alter our adherence to PLOS ONE policies on sharing data and materials.” (as detailed online in our guide for authors hkp://journals.plos.org/plosone/s/competing-interests) . If this adherence statement is not accurate and there are restrictions on sharing of data and/or materials, please state these. Please note that we cannot proceed with consideration of your article until this information has been declared.

*****Not applicable because there was no commercial affiliation, but only the intellectual contribution of someone acting completely independently of Pupa. *****

*****Nothing to update. *****

Please know it is PLOS ONE policy for corresponding authors to declare, on behalf of all authors, all potential competing interests for the purposes of transparency. PLOS deﬁnes a competing interest as anything that interferes with, or could reasonably be perceived as interfering with, the full and objective presentation, peer review, editorial decision-making, or publication of research or non-research articles submitted to one of the journals. Competing interests can be ﬁnancial or non-ﬁnancial, professional, or personal. Competing interests can arise in relationship to an organization or another person. Please follow this link to our website for more details on competing interests: hkp://journals.plos.org/plosone/s/competing-interests

*****There are no competing interests. ***** 

Additional Editor Comments (if provided): 

The Authors are expected to address all the criticisms by all Reviewers. In particular, avoid implication on causality, justify, or avoid exclusion of countries, including countries in the Southern Hemisphere (Reviewers #2 and #3), improve the writing of the manuscript, revise Figure 2 for accuracy and clarify (Reviewer #1), assess the impact of the choice of time period on the results (Reviewer #2), avoid conclusion on seasonality, and provide the equations for the major models used for the analysis (Reviewer #3). 

*****All of these points were addressed in responses to reviewers below. *****

In additional to the above comments, please address,

1. L87-90, the exclusion of China, Russia and Italy is not well justiﬁed. The case numbers in diﬀerent countries are available from diﬀerent sources which were translated into English. There are other exclusions which were not fully justiﬁed.

*****Justifications for inclusions and exclusions were provided at length (see below where reviewer raised this concern, Lines 402-447). *****

2. Please remove citations 6 and 7

*****Citations 6 & 7 were removed and references from the published, peer-reviewed literature substituted instead. *****

3. Carry out relevant sensitivity analysis to justify the conclusion that the results were robust.

*****See sensitivity analysis for 1st regression and for temperature (lines 322-358) *****

Reviewers' comments:

Reviewer's Responses to Questions

Comments to the Author

1. Is the manuscript technically sound, and do the data support the conclusions?

The manuscript must describe a technically sound piece of scientiﬁc research with data that supports the conclusions. Experiments must have been conducted rigorously, with appropriate controls, replication, and sample sizes. The conclusions must be drawn appropriately based on the data presented.

Reviewer #1: No

Reviewer #2: Partly

Reviewer #3: Partly

2. Has the statistical analysis been performed appropriately and rigorously? Reviewer #1: No

Reviewer #2: No

Reviewer #3: Yes

3. Have the authors made all data underlying the ﬁndings in their manuscript fully available?

The PLOS Data policy requires authors to make all data underlying the ﬁndings described in their manuscript fully available without restriction, with rare exception (please refer to the Data Availability Statement in the manuscript PDF ﬁle). The data should be provided as part of the manuscript or its supporting information, or deposited to a public repository. For example, in addition to summary statistics, the data points behind means, medians and variance measures should be available. If there are restrictions on publicly sharing data—e.g. participant privacy or use of data from a third party—those must be speciﬁed.

Reviewer #1: No

Reviewer #2: Yes

Reviewer #3: Yes

4. Is the manuscript presented in an intelligible fashion and written in standard English?

PLOS ONE does not copyedit accepted manuscripts, so the language in submitted articles must be clear, correct, and unambiguous. Any typographical or grammatical errors should be corrected at revision, so please note any speciﬁc errors here.

Reviewer #1: No

Reviewer #2: Yes

Reviewer #3: Yes

5. Review Comments to the Author

Reviewer #1: If only it were that simple! On the whole, linear regressions can generate hypotheses, but they can't test them in the way that you are attempting to do. 

*****In fact the use of linear regression to understanding epidemiological factors influencing the transmission dynamics of viral transmission has a long history in the literature. Let it suffice for us here to provide three examples where linear regression was applied in a similar fashion to evaluate factors associated with SARS-CoV-2 transmission, the same context for our study. (3)(4)(5) *****

Figure 2 looks very impressive until you realize that it is extracted from a model where you've got quite a few variables, and not that many data points. The correlation shown is - or seems to be - meaningful and important. The raw and derived data (including your CT measure) is much less impressive.

I may have misunderstood Figure 2, but then it is very poorly explained – see my comments below.

I hesitate to criticize you for this, because a lot of well-known epidemiologists are guilty of the same thing, but it would help a lot if you were clearer about the phenomena that you are modelling, and what we don’t know.

Epidemiologists use very simple models where they assume that someone who is infected always becomes symptomatic after a certain interval. This is in reality not the case. A very interesting series of articles were published recently by Jeﬀ Shaman’s group at Columbia University who sampled the population of New York City, looking for the presence of respiratory viruses. They found that the majority of people carrying detectable virus had no symptoms. They also found that as many people had detectable virus in summer as in winter. I feel your article (or similar articles in future) would be improved if you thought more about the possible mechanisms that give rise to seasonality, and were clearer about what we DON’T know. I’ve put some references below.

*****It appears that the reviewer is suggesting that models that incorporate consideration of the contribution of asymptomatic viral carriers to transmission are worthy of consideration in the study of COVID-19 dynamics. While theoretically of interest, and quite likely a worthy hypothesis warranting investigation by the reviewer if he is so motivated, it has little bearing on our study. Our investigation of the association of temperature and rate of viral transmission holds regardless of such considerations, assuming that there are no systematic regional differences in the rate of asymptomatic carriers--for which there is no evidence to suggest such a difference exists. *****

Can I ask that in future you take a little more trouble in writing your articles? There are lots of problems with this manuscript. For a start, there are many sentences that don't have verbs in them. For example, p8, line 119. In the abstract, P2, lines 44-45 don't make sense. Then on the most important item in your paper, Figure 2, the vertical axis is not labelled correctly. In fact Fig 2 says that as temp increases the number of cases increases! You are also inconsistent in abbreviations - sometimes you write DP, other times DEWP. It's not fair on the reviewers to ask them to put in a lot of eﬀort if you don't make your article easy to read.

*****We greatly appreciate the reviewers identifying, so we could correct them, the three written mistakes in the entire body of the manuscript. We would add parenthetically, that three mistakes in our humble opinion do not rise to the level of “lots of problems.”

a. The inclusion of the words “are shown” was made to p8, line 119. 

b. The word “is” was inserted into P2, line 44 for clarification. 

c. The single use of the abbreviation DEWP in Table 1 was corrected to fit with the other 21 correct uses of the abbreviation DP

We also appreciate the reviewer identifying our mislabeling of the Y axis in Figure 2 with the same label used in Figure 1. Although the correct information was provided in the figure legend, this error was a mistake that made the interpretation of the figure unnecessary confusing. We have changed the Y axis to be the correct information as shown in the figure legend. 

We would add parenthetically that Fig 2 is not, as the reviewer suggests, “the most important item” in our paper. Quite the contrary, when the manuscript was reviewed but ultimately passed on at JAMA we had not included Figure 2 at all. We only added it to our PLOS ONE submission to lend a visual confirmation of the far more important statistical analysis results shown in Table 2. Most multilinear regressions in the literature (see references of other manuscripts employing regressions to characterize SARS-CoV-2, above) do not go through the trouble of showing such a graphical representation. *****

There is also a lot of repetition in the article between the table and ﬁgure legends and the main text. Just write everything once, clearly. I always start by writing the legends, because they are what most people read ﬁrst. Either put the detail in there, or refer, in the legends, to the main text for the detail. Most people write the main text ﬁrst, then they realize that they need to explain the ﬁgures, and write the same material over again - I think this might be what you have done.

*****We agree wholeheartedly with the reviewer, and always start from the identification of the tables and figures that will comprise the “story” we intend to “tell” in our manuscript, and then let the main text follow. *****

Galanti, M., et al. "Rates of asymptomatic respiratory virus infection across age groups." Epidemiology & Infection 147 (2019).

Galanti, M, et al. "Longitudinal active sampling for respiratory viral infections across age groups." Inﬂuenza and Other Respiratory Viruses 13.3 (2019): 226-232.

Shaw Stewart, PD. Seasonality and selective trends in viral acute respiratory tract infections. Medical Hypotheses 2016; 86 104–119. hkps://www.douglas.co.uk/f_\\p1/ShawStewart_ﬁnal_1-s2.pdf

Reviewer #2: The paper is of interest and I would like to see a number of improvements to this work before it can be published.

The paper looks at the association between the COVID-19 case number and the temperature. This topic is not novel and has been widely studied (e.g., [1-5ti). However, the current manuscript does not give a clear and comprehensive review of the most up-to-date background and studies of this topic. The author should clearly address the innovative contribution of their study and compare with other relevant studies (e.g., whether a new method is used? what is the advantage of this new method? Does the study cover a more comprehensive dataset? Does the study give any new conclusion or verify existing results?) These should be discussed in the Introduction and Discussion of this manuscript.

*****We greatly appreciate the reviewers recommendation that we include a clear and comprehensive review of the most up to date background studies (which included all of the references he provided) and a discussion of what was innovative about our approach. We included the review in the introduction (lines 71-84)and the discussion of the innovative contribution of our work in the discussion (lines 455-471). *****

In the Method section, the authors state that they exclude data from China and Russia and list the relevant news report as a support. It is generally acceptable to cite information from the media for the COVID study when the information is simply an objective report of what has happened. However, the citation 6 and 7 in the manuscript are based on some so called “source of the leak” and cannot be veriﬁed. It is just a speculation made by someone not from the scientiﬁc community. The WHO has never made a similar statement. Therefore, it is not scientiﬁc to exclude data from China and Russia based on current “support”. The authors should cite reliable and objective information (e.g., published papers, as the author already do to explain Italy) or give some more convincible reasons. Similarly, I am interested to know the reason of excluding Canada and the United States of America.

*****We appreciate the chance to elaborate further, and improve our references to recently published peer reviewed journals, related to the reasons for exclusion of China, Russia, Canada and the US. To give an appropriate background, we believe among the reasons we were able to tease apart such a strong finding of the impact of temperature on SARS-CoV-2 transmission (with an R^2 of 0.6 and a p<0.000005) involves our systematic use of the information from the COVID-19 Data Repository by the Center for Systems Science and Engineering (CSSE) at Johns Hopkins University (https://github.com/CSSEGISandData/COVID-19). The data in this repository lists daily cumulative cases by country, and provides a latitude and longitude for each country. To gather the most relevant meteorological data for each country, we identified the weather station via NOAA website (https://www7.ncdc.noaa.gov/CDO/cdoselect.cmd?datasetabbv=GSOD&countryabbv=&georegionabbv=) that was closest to the latitude and longitude provided by the JHU CSSE database--to ensure as close a correlation between the epicenter of the countries COVID-19 data and the actual weather in proximity to that location. Now to the exclusion reasons for China, Russia, the US and Canada. 

1. We excluded countries with multiple data sites (for both COVID-19 cases and NOAA meteorological data) within its borders. Each country had differing responses to the increasing rates of COVID-19 within its borders (e.g. the timing and availability of testing, government policy decisions about when and how robustly to respond with non-pharmaceutical interventions). Because each country that was included contributed data representing a single response to COVID-19, the variability in the response would be randomized (some countries more and some less in various aspects of their response). The inclusion of a single country with numerous measures within its borders would risk creating a systematic bias for that country in terms of a potential effect on the rate of transmission of COVID-19. The following is a list of the number of separate locations with reported COVID-19 cases: US 3340, China 33, Canada 14. Thus, the US, China and Canada were excluded based on this concern. 

2. Russia presented the problem that a country of that magnitude had only a single JHU CSSE COVID-19 data entry with a single latitude and longitude, which could clearly not accurately represent the appropriate temperature corresponding to the rate of increase of cases for the entire country.

3. Another reason why Canada and Russia were excluded related to their land area per capita (LAPC), which had an impact on the multivariable regression to establish the correlation between median temperature and rate of increase of cases. Whereas for the 50 countries included, the average and median LAPC were 0.02 & 0.009, respectively, the LAPC for China was 0.24 (26 vs 12 fold greater than the average and median of the 50 countries) and Russia 0.11 (12 and 6 fold greater). In effect, the relatively low population density in these countries would be expected to significantly influence (i.e. slow down) the rate of increase in COVID-19 cases and make these outliers in the data analysis. 

4. A further reason to exclude China and the US relate to their level of under-reporting of cases. A recent publication by Krantz et al(6) that reported the ratio of reported COVID-19 cases to model-based predictions of COVID-19 for 8 major countries. The ratio of reported to infected cases for China was between 1:149 to 1:1104, and for the US was 1:504. In comparison, France was 1:5 and Germany was 1:4. 

5. A final reason for not using China is that the one study that showed no effect of temperature on SARS-CoV-2 transmission was done using available reports from China (7), which conflicts with multiple other reports demonstrating some effect of temperature on viral transmission. This further supports the atypical nature of COVID-19 that is unique to studies of China, for whatever reason (e.g. inaccurate data or uniquely unusual viral behavior in this country of origin of the Pandemic). *****

The authors state that the method used in this study is robust. By looking at the Figure 1, I doubt if the conclusion made in the manuscript is really robust. The cumulative number of COVID 19 cases generally shows a concave trend. Hence, if you look at the ﬁrst half of the time period, the coeﬃcients of time (CT) will be larger. Similarly, the CT will be smaller if you look at the other half of the time period. It is therefore needed to investigate whether the conclusion still holds when the time period changes. If the conclusion is largely aﬀected, this point should be stated as a caution in the manuscript. Meanwhile, the author may consider including the quadratic term in the ﬁrst regression.

*****The reviewer has a keen eye in observing the concave trend of the curve over time. And we appreciate the opportunity to explain why we analyzed the initial phase of the increase in SARS-CoV-2 :

1. The initial rise in the rate of SARS-CoV-2 is linear as evidenced by a mean R^2 of 0.91. This is the phase where the virus is beginning its spread through a country. During this early phase there is a delayed response on the part of the country (usually by both the government and the populace) to institute and practice non-pharmaceutical interventions (NPI) to slow the spread of the virus.

2. With time countries begin to respond to the spreading pandemic (see https://www.bsg.ox.ac.uk/sites/default/files/2020-09/BSG-WP-2020-032-v7.0.pdf for details of the timing of country level responses). These NPIs are geared to “bend the curve” of progression (i.e. slow the increase in the rate of transmission). The speed of government response of each country, and the rate of success of citizens at implementing NPI, varies widely introducing a significant variable in addition to any impact of weather on the rate of transmission. 

3. As an example of the impact of NPI on the concave curvature of cumulative number of cases, see the accompanying graph titled “NYC Lock Down” immediately below. Because of the incredibly rapid rate of spread of the pandemic in NYC (which had cold temperatures at the time), a strictly enforced and adhered to NPI (referred to as a “lockdown”) was instituted. It is possible to see clearly the change in rate of new cases after the lockdown in NYC. To demonstrate the change was specific to the lockdown, we have included a comparison with Poland where no such dramatic interventions were instituted.

4. By focusing our analysis on the initial linear segment of rate of accumulated COVID-19 cases we were able to isolate the impact of weather from the impact of social interventions (i.e. NPI). That is in part why we believe we were able to resolve such a clear and unambiguous signal across the norther hemisphere in 50 representative countries. 

5. It is worth noting that we believe that it should be possible, perhaps in a follow-up project, to resolve the contribution of NPI on the bending of the curve. In the absence of a vaccine, the actual rate of decrease of transmission equals the rate of NPI influence plus the rate of temperature effects, one should be able to subtract the calculated impact of temperature (e.g. from measures such as our manuscript) from the observed changes to quantitate the effectiveness of NPI. 

Regarding the robustness of the findings, at the excellent suggestion of the reviewer we performed a sensitivity analysis by changing the time period which was sampled. There was no meaningful change to the results, and this analysis has been added to the manuscript to reassure the reviewer (see Lines 322-358) *****

The authors should be cautious when saying “a one degree increase leads to a 1% decrease--and a one degree decrease leads to a 3.7% increase” in the Abstract. “Lead” normally indicates causality. It should be replaced by “associate”.

*****We appreciate the reviewers catching our overzealousness in lines 47 and 48, where we used the words “leads to” instead of “is associated with.” We corrected this error, and further searched the rest of the manuscript where we were reassured that we used the term “associated” and not “leads” throughout the rest of the submission. *****

Minor Comments:

In line 57, it should be “Coronavirus” rather than “Corona VIrus”

*****We greatly appreciate the reviewers time and detailed editing of our manuscript, and have made the change he recommended. We had intended the capitalization of the letters to correspond to the acronym COVID, but it was clearly confusing for us to do so. *****

In Figure 2, the y axis label should be “reciprocal of CT (CT-1)” rather than “conﬁrmed cases (Log). 

*****We greatly appreciate the reviewer’s noticing our mistake in carrying over the Y axis label from figure 1 to figure 2. This greatly confuses the figure and what it demonstrates. We made the change, which spares future readers extra work ,and us extra embarrassment, in appreciating the significance of the figure. *****

Additional Reference:

1. Wu Y, Jing W, Liu J, et al. Eﬀects of temperature and humidity on the daily new cases and new deaths of COVID-19 in 166 countries. Sci Total Environ. 2020;729:139051. doi:10.1016/j.scitotenv.2020.139051

2. Shi P, Dong Y, Yan H, et al. Impact of temperature on the dynamics of the COVID-19 outbreak in China. Sci Total Environ. 2020;728:138890. doi:10.1016/j.scitotenv.2020.138890

3. Xie J, Zhu Y. Association between ambient temperature and COVID-19 infection in 122 cities from China. Sci Total Environ. 2020;724:138201. doi:10.1016/j.scitotenv.2020.138201

4. Ujiie M, Tsuzuki S, Ohmagari N. Eﬀect of temperature on the infectivity of COVID-19. Int J Infect Dis. 2020;95:301- 303. doi:10.1016/j.ijid.2020.04.068

5. Sajadi MM, Habibzadeh P, Vintzileos A, Shokouhi S, Miralles-Wilhelm F, Amoroso A. Temperature, Humidity, and Latitude Analysis to Estimate Potential Spread and Seasonality of Coronavirus Disease 2019 (COVID-19). JAMA Netw Open. 2020;3(6):e2011834. Published 2020 Jun 1. doi:10.1001/jamanetworkopen.2020.11834

Reviewer #3: Thank you for giving me an opportunity to review this submission titled “Evidence and magnitude of seasonality in SARS-CoV2 transmission: Penny wise, pandemic foolish?”. The study is very timely given the current pandemic situation and indeed tackles an important issue, and to some extent controversy, about environmental and atmospheric conditions conducive for COVID-19 transmission. I acknowledge that the article has its own merits and immense potential to be able to contribute to current body of knowledge; however, there are still quite a few issues I found perplexing and needs to be addressed better before I give my full recommendation for publication. The ﬁrst thing that stands out is the use of the word “seasonality” which to me might be a little bit misleading as eﬀectively only one season was evaluated. It is a strong claim to make given very limited evidence presented in the study and it did not help that the design further limited this viewpoint by not considering Southern Hemisphere countries/regions. Rather, a more appropriate wording must have focused around correlation between meteorological or environmental or atmospheric conditions with COVID-19 transmission. I have some more speciﬁc questions and concerns detailed below. Pending satisfactory responses, I can then give my recommendation to have this published.

*****We greatly appreciate the reviewers kind words about the merits and immense potential of our manuscript, and further are grateful for her constructive critique to aid in the improvement of its shortcomings.

To that end, as suggested by the reviewer, we changed the title to: “Evidence and magnitude of meteorological changes on SARS-CoV2 transmission” *****

Abstract:

• Line 38: Can you expound more, either in the Methods or Results section, how the reciprocal of CT can be associated with doubling time? If possible, is there some exact mathematics as to how they are related?

*****We greatly appreciate the reviewer providing this fascinating question. We ran empiric calculations of the Doubling Tim (Td) for each country and then ran multiple linear regression to investigate the relationship between Td (doubling time) and CT. We included all of this in our manuscript and believe the reviewers recommendation to expand on this point has enhanced our manuscript significantly. See lines 388-414. *****

• Lines 49-50: While it does make sense to make the claim about summer months, I ﬁnd it hard to make a claim on winter months given there is no actual evidence to support this. Data were limited to the Northern Hemisphere countries which are yet to experience winter, and all that we have are claims based on temperature changes which is not fully indicative of a seasonal change. Actually, this applies to the summer claim as well. Perhaps, this statement can be restated so that it does not appear to be overpromising and just stick with correlations between temperature or humidity with transmission?

*****The reviewer is of course correct, and makes an important point—from the data shown it is only possible to definitively conclude the association of increasing temperature with decreasing rates of transmission. 

1. Because of this we have changed the wording from “with the reverse true in winter months” to “consistent with the behavior of a seasonal respiratory virus.” Thus, we changed the claim from conjecture into a statement of fact.

We suggest to the reviewer that there are several reasons to utilize what is already known about the biology of respiratory viruses like SARS-CoV-2 in the interpretation of our findings—i.e. suggesting seasonality--for the following reasons:

2. Seasonality in respiratory viruses is characterized by decreased transmission in warmer and more humid months, and increased transmission in cooler, less humid months. It has been demonstrated that for respiratory viruses, including Influenza viruses as well as SARS-CoV-1 and SARS-CoV-2, the stability of the envelope surrounding and protecting these viruses is temperature sensitive—degrading more rapidly at higher temperatures and showing increased stability at cooler temperatures in laboratory studies. This is thought to be one of two key factor related to the mechanism by which these viruses manifest their seasonality, the other being humidity (where lower humidity leads to smaller droplets that are more volatile and capable of lodging deeper in the lung). 

3. SARS-CoV-1, which shows seasonal changes with decreased rates of transmission with elevated temperatures, is genetically 80% identical to SARS-CoV-2. 

4. The results of our manuscript, we suggest, can be viewed in the context of the biology of coronaviruses in general--including SARS-CoV-1 that is closely related to SARS-CoV-2-- and laboratory evidence for SARS-CoV-2 specifically showing decreased viral stability at higher temperatures and increased stability at lower temperatures. The data from this paper, when viewed in this context, we submit is strongly suggestive of the temperature sensitivity of transmission rates being a manifestation of the seasonal changes known from biological and epidemiological studies of betacoronaviruses in general and SARS-CoV-2 specifically. 

5. As explained below, in our response to the first point in Methods section of this reviewers critique, we could not include data from southern hemisphere countries and hope to achieve a clear signal of the association of temperature on rates of transmission. 

6. Finally, the fact that this data, when viewed in its biological and epidemiological context, suggests seasonality is a key component of what the reviewer rightly points out to be one of the important implications of this research. The reviewer states in her introduction: “The study is very timely given the current pandemic situation and indeed tackles an important issue, and to some extent controversy, about environmental and atmospheric conditions conducive for COVID-19 transmission.” The only way anyone will know with certainty that the SARS-CoV-2 rates of transmission will go up in cooler temperatures (as happens with all coronaviruses that show decreases with warmer temperatures) is to wait and see what happens over the fall and winter. But the discussion of the possible implications of temperature sensitivity that we demonstrate in warmer temperatures to the rates of transmission in cooler months is precisely what is needed for our findings to “tackle an important issue.” And only by raising this possibility can we hope to have the potential to save lives in the immediate future, by warning of the potential for increased transmission in the next few months. For us, we think the importance of potentially saving lives outweighs our need to wait until there is definitive proof of the seasonality of SAR-CoV-2.

That being said, and following the reviewers recommendation, we suggested that an increase in transmission rate “could be” (changed from “is”) and clarified that this is only a proposed model rather than a direct finding of our study: “This model of the effect of decreasing temperatures can only be verified over time as the pandemic proceeds through colder months.” (See lines 50-55) *****

Methods:

• Lines 84-87: However, my take is that wouldn't using Southern Hemisphere countries even as controls be also helpful, as at the same time they experienced drop in temperatures? Especially, if the study is also making a claim about possible 'resurgence' during colder months? In fact, I am not convinced why does the data cannot from two hemispheres cannot be pooled.

*****The reviewer raises a very good point and we were tempted to include Southern Hemisphere countries at the start, but we were concerned about the possible asymmetry of the effects of changing temperatures on the increasing vs decreasing rates of transmission as described below. Here is why we believe our approach to analyzing the association of the rate transmission with temperature was best limited to the northern hemisphere:

1. To begin with, both the origin of transmission in the northern hemisphere, and the rates of global spread of COVDI-19 faster in the northern latitudes than the southern, meant that the data available for southern hemisphere countries was significantly restricted over the time period we sampled (especially when one considered the trailing rates of deaths). 

2. Sampling from a later time from Southern latitudes would be further complicated by the fact that lessons learned by observing the spread in Northern neighbors could have exerted a different response in the Southern countries, thereby affecting early rates of transmission. *****

• Lines 87-90: While not so much of an issue at the moment, but have you considered exploring or conducting some kind of sensitivity analyses to check how the results would have changed if these countries are added in the analyses– at least even just with Italy?

*****We appreciate the chance to elaborate further, and improve our references to recently published peer reviewed journals, related to the reasons for exclusion of China, Russia, Canada and the US. To give an appropriate background, we believe among the reasons we were able to tease apart such a strong finding of the impact of temperature on SARS-CoV-2 transmission (with an R^2 of 0.6 and a p<0.000005) involves our systematic use of the information from the COVID-19 Data Repository by the Center for Systems Science and Engineering (CSSE) at Johns Hopkins University (https://github.com/CSSEGISandData/COVID-19). The data in this repository lists daily cumulative cases by country, and provides a latitude and longitude for each country. To gather the most relevant meteorological data for each country, we identified the weather station via NOAA website (https://www7.ncdc.noaa.gov/CDO/cdoselect.cmd?datasetabbv=GSOD&countryabbv=&georegionabbv=) that was closest to the latitude and longitude provided by the JHU CSSE database--to ensure as close a correlation between the epicenter of the countries COVID-19 data and the actual weather in proximity to that location. Now to the exclusion reasons for China, Russia, the US and Canada. 

1. We excluded countries with multiple data sites (for both COVID-19 cases and NOAA meteorological data) within its borders. Each country had differing responses to the increasing rates of COVID-19 within its borders (e.g. the timing and availability of testing, government policy decisions about when and how robustly to respond with non-pharmaceutical interventions). Because each country that was included contributed data representing a single response to COVID-19, the variability in the response would be randomized (some countries more and some less in various aspects of their response). The inclusion of a single country with numerous measures within its borders would risk creating a systematic bias for that country in terms of a potential effect of their response to COVID-19. The following is a list of the number of separate locations with reported COVID-19 cases: US 3340, China 33, Canada 14, Russia 1. Thus, the US, China and Canada were excluded based on this concern. 

2. Another reason why Canada and Russia were excluded related to their land area per capita (LAPC), which had an impact on the multivariate regression to establish the correlation between median temperature and rate of increase of cases. Whereas for the 50 countries included the average and median LAPC were 0.02 & 0.009, respectively, for the LAPC for China was 0.24 (26 vs 12 fold greater than the average and median of the 50 countries) and Russia 0.11 (12 and 6 fold greater). In effect, the relatively low population density in these countries would be expected to significantly influence (i.e. slow down) the rate of increase in COVID-19 cases. 

3. Russia also presented the additional problem that a country of that magnitude had only a single JHU CSSE COVID-19 data entry with a single latitude and longitude, which could clearly not accurately represent the appropriate temperature corresponding to the rate of increase of cases.

4. A further reason to exclude China and the US relate to their level of under-reporting of cases. A recent publication by Krantz et al(6) that reported the ratio of reported COVID-19 cases to model-based predictions of COVID-19 for 8 major countries. The ratio of reported to infected cases for China was between 1:149 to 1:1104, and for the US was 1:504. In comparison, France was 1:5 and Germany was 1:4. 

5. A final reason for not using China is that the one study that showed no effect of temperature on SARS-CoV-2 transmission was done using available reports from China (7), which conflicts with multiple other reports demonstrating some effect of temperature on viral transmission. This further supports the atypical nature of COVID-19 that is unique to studies of China, for whatever reason (e.g. inaccurate data or uniquely unusual viral behavior in this country of origin of the Pandemic). 

At the excellent suggestion of this reviewer, we conducted a sensitivity analysis to assess the impact of changing the dates over which cases and temperatures were calculated. We noted that changes as much as 22% earlier or 22% later had no significant impact on the major findings of our study. (see lines 325-376) *****

• Lines 91-92: What did you mean by countries with separate reports of cases from multiple regions? Can you list what these countries are?

*****See point #1 in the preceding section related to our justification of country inclusion and exclusion criteria. *****

• Line 93: How did you assess reliability of COVID-19 and meteorological data?

*****See points #3 and #4 for how our selection criteria above (lines 615-626) excluded countries with known reliability problems in COVIDI-19 cases. Regarding meteorological data, it is collected by NOAA stations throughout the world in a systematic manner with comparable precision. *****

• Lines 107-111: Were there opportunities to also used other variables? While it does help that you provided a reference as to why Tmin is one of the most appropriate to use in this context, I would be inclined to also explore other measures (such as Tmean, Tmed, Tmax) and see how the correlation magnitudes change or not. To my mind, this could have presented a more comprehensive picture of how temperature is correlated with transmission. This also applies to other atmospheric data which NOAA is tracking and ‘complete’ for all countries analysed.

*****These are excellent questions. We included in our analysis a host of variables, including all of the NOAA tracked metrological variables and only those that had a statistically significant impact on the multiple linear regression were included in the manuscript. These were the NOAA variables included in the multivariable regression: 

Average Dew Point, Maximum Wind Gust, Maximum Temperature , Minimum Temperature , Maximum Sustained Wind Speed , Precipitation , Average Sea Level Pressure , Snow Depth , Average Station Pressure , Average Temperature , Average Visibility , Average Wind Speed. *****

• Lines 113-114: I’m sorry but I did not understand this fully. So does this mean whenever there were 0 cases in a day, corresponding Tmin and DP were not also recorded? Why is this so? Even if, say, there is one day that no conﬁrmed cases were reported squeezed between two days with high cases, that day with 0 case is already dropped? But I still don’t see why this is needed to be performed given it is the reciprocal of CT (based on cumulative counts) that is being modelled. So in other words, the entire span of the series for CT per country should also be the same timeline of the temperature and dewpoint data to be considered. Or maybe this is already performed, but I am just understanding this incorrectly? Can you please explain further? Best if you can give me an actual scenario/example?

*****We apologize if have not understood the question the reviewer is posing. In our version of the submission, 113-114 does not make mention of 0 cases in a day. We wonder whether the reviewer is commenting on this sentence from the manuscript (lines 99-103): 

1. “To enable analysis of cases during consecutive days in a log linear fashion, we established the following criteria to ensure sufficient numbers of confirmed cases: to be included, during time period 1/22/20-4/6/20 countries needed to have a median number of confirmed cases >1 or the mean number of confirmed cases >75.” 

2. Here are some of the examples of countries that would not meet this criteria (name followed by sequential daily cumulative number of confirmed cases from 1/22/20-4/6/20):

a. Bhutan 0 0 0 0 0 0 0 0 0 0 0 0 0 0 0 0 0 0 0 0 0 0 0 0 0 0 0 0 0 0 0 0 0 0 0 0 0 0 0 0 0 0 0 0 1 1 1 1 1 1 1 1 1 1 1 1 1 1 2 2 2 2 2 2 2 3 3 4 4 4 4 5 5 5 5 5

b. Nepal 0 0 0 1 1 1 1 1 1 1 1 1 1 1 1 1 1 1 1 1 1 1 1 1 1 1 1 1 1 1 1 1 1 1 1 1 1 1 1 1 1 1 1 1 1 1 1 1 1 1 1 1 1 1 1 1 1 1 1 1 1 2 2 3 3 4 5 5 5 5 5 6 6 9 9 9

c. Laos 0 0 0 0 0 0 0 0 0 0 0 0 0 0 0 0 0 0 0 0 0 0 0 0 0 0 0 0 0 0 0 0 0 0 0 0 0 0 0 0 0 0 0 0 0 0 0 0 0 0 0 0 0 0 0 0 0 0 0 0 0 0 2 3 6 6 8 8 8 9 10 10 10 10 11 12

3. We believed due to the low number of cases in these countries there would not be a sufficient amount of data to extrapolate whether there was an impact of temperature on transmission. *****

• Can you add a subsection describing with thorough details the model formulations (including even just ‘generic’ equations but appropriately contextualised)?

*****We inserted a subsection titled “Details the model formulations (including equations in appropriate context)”Lines 443-433. *****

Results:

• Lines 150-152: Are there other measures you can consider to assess the goodness-of-ﬁt? To my mind, while R2 is okay, we will actually expect it to be very high given we are ﬁtting to a function of cumulative conﬁrmed cases which are naturally increasing in in time. So no matter what the shape is, especially with log(cumulative cases), the linear increasing trend will be captured very well. I will be more inclined to report other goodness-of-ﬁt measures too just as an added support to this great ﬁt.

*****We performed a sensitivity analysis to test whether our findings are robust. We found no significant changes in our two step regression results even if we vary the timeline during which cases accumulated and temperatures were calculated. See lines 352-378. *****

• Figure 2: It seems to me that there is a systematic bias here, where for higher values the best-ﬁt (predicted) values are always less than those? In other words, for higher conﬁrmed inverse CT there appears to be a systematic underestimation. Wouldn’t this aﬀect model interpretation and appropriateness for higher values of either inverse CT or temperature?

*****The reviewer has a very keen eye! The cause of what appears to be a relatively small systematic underestimation for higher confirmed CT^-1 is actually due to changes at the lower end of the curve: below freezing there is a gradual leveling off of the decrease in confirmed cases, presumably because once frozen progressively lower temperatures no longer facilitate coronavirus transmission. The effect of this levelling off below freezing on linear regression fitting shown in the figure results in what appears to be an under-estimation at higher temperatures. *****

• Lines 224-225: How often do you see this much change in temperature (drop or increase of 70°F)? To me, I would prefer to present them in a much more interpretable format, say the relative size of temperature changes going from one latitudinal zone to another. Or even simpler, in around 5-10°F changes?

*****The reviewer raises an excellent point—we had not sited why we chose 30-70 degree change. We had taken the change in temperature from the following graph of Northern Hemisphere Seasonal Changes (-2 to 21 Celsius translates to 28.4 to 69.8 F):

http://berkeleyearth.lbl.gov/regions/northern-hemisphere

That being said, if the reviewer prefers us to us the minimum and maximum Tmin_Med from the 50 countries we included, which is 15.5 and 77.4 respectively, we will of course defer to her judgement. Here is what the paragraph would state if we changed to 15.5 and 77.4 F:

“Magnitude of Effect of Temperature Change on Rate of Transmission of COVID-19:

The minimum and maximum and Tmin_Med in the 50 representative countries examined was 15.5 and 77.4 °F, respectively (see Table 1). From the second regression described above, the following equation was derived:

CT = 0.116 - 0.000911 x Tmin_Median

It was therefore possible to obtain CT for temperatures ranging from 15.5 °F to 77.4 °F – i.e. 0.1018 (log cases/day) to 0.0454 (log cases/day), respectively. This analysis indicated that an increase in temperature from 15.5 to77.4 °F was associated with a 55% decrease in the rate of number of confirmed cases per day (CT 0.1018 to 0.0454 ), whereas a decrease in temperature from 77.4° to 15.5 °F represents a 124 % increase in the rate of change of confirmed cases (CT 0.0454 to 0.1018 ). This corresponds to 9% decrease in confirmed cases for each 10 degree increase in temperature, and a 20 % increase for each 10 degree decrease in temperature.”

Italicized text represents the changes. We also added the final sentence about the change in rate of confirmed cases for a 10 degree change in temperature. *****

Discussion:

• Line 240: How do you qualify ‘strong and robust’ with your ﬁndings? To me, I feel that this is such a strong claim and a bit of an overstretch given the issues I pointed out earlier about data catchment and quality as well as standard correlational approaches via regression. Perhaps, consider rephrasing this so as not to oversell the results?

*****Please see our response to the critique you provided in Lines 150-152 above. As a result of your very helpful suggestion we have incorporated in our manuscript a sensitivity analysis that demonstrates the robust aspect of the finding. We removed the word “strong.” See lines 325-376. *****

• Line 242: Again, I ﬁnd ‘impacts’ a bit too strong given there is no causality established - what we only know is that there is statistical evidence (from correlations) that they are related.

*****We have followed the reviewers suggestion and changed “impacts” to “associated with” as follows:

“The results of our analysis of longitudinal data from 50 representative countries in the Northern Hemisphere suggest that changes in the minimum daily temperature are associated with alterations in SARS CoV-2 transmissibility and viability.” Lines 446-449. *****

• Line 272: In claiming similarities in seasonal pattern of infections with SARS-CoV, perhaps you can add more context into this. As your study focused only on temperature (and to some extent, humidity) perhaps it is better if you discuss this in the context of changes in temperature (or ranges) here from summer to winter. Again, it is hard to make claims about summer and winter diﬀerences when all your data are based on summer and the only proxy for winter conditions is based on temperature. Furthermore, I suppose this seasonal pattern can be diﬀerent in many regions, and what happens for tropical climates where there is no winter?

*****Please see our response to this important issue that the reviewer raises about winter vs summer interpretations above related to the critique raised in Lines 49-50. We also appreciate the issue of the lack of change in temperature in tropical climates, which we neglected to add, so we have inserted this sentence into the discussion of regions included or excluded from our consideration: “We excluded tropical regions because of the minimal change in temperature throughout the year.” *****

• Lines 293-297: Can you further explain this sentence? Of course, they will be diﬀerent from a calculation perspective given you have diﬀerent denominators in quantifying changes between increasing-to-decreasing and decreasing-to-increasing even for the same magnitude (70°F) of diﬀerence. To my mind, they are all based on the estimated regression coeﬃcients so in principle the changes should be similar.

*****Please see our response to the reviewers points raised about Lines 224-225. Moreover, the point we are trying to make is that if a Northern Hemisphere country permits the same amount of cases to be present as temperature drops as they do when temperature rises, the increase in transmission with dropping temperature will outpace the decrease seen as temperature increases. This asymmetry is similar to how in the stock market, if one loses 50% of the price of your stock one year, then would have to achieve a 100% rise in the same stock to recoup your losses. *****

• Line 299: Again, I ﬁnd the claim that “could pay oﬀ signiﬁcantly in the fall and winter” a bit of an overstretch given the same issues I raised before about not having analysed winter conditions (e.g., by not considering the Southern Hemisphere).

*****Please see our response to this important topic you raised above in Lines 49-50. *****

Some minor comments:

• In general, when presenting temperatures, would you consider mentioning them in °C too, especially for audience who use this scale? This adds more context in the study.

*****Because the range of °F is 180% broader than °C—i.e. where from freezing to boiling is 180 degrees °F as opposed to 100 degrees C, respectively--°F proved best for capturing the changes in temperature and in the precision it provided. We also believed that following our analysis, converting results from °F to °C should be relatively easy, and would not necessitate an elaborate explanation for why the findings (e.g. R^2, p values) etc were different between the two systems of measure (°F vs °C). *****

• Also, maybe reconsider restructuring the Methods and Results sections, as some contents are better written under Methods (e.g. model-ﬁttng stages) while others are in Results (e.g. Table 1).

*****We tried our best to “tell a story” by presenting the information in which a reader could best follow the steps taken and the reasoning employed throughout our endeavor. We agree with the reviewer that Table 1, which has both the Methods component of listing the countries selected and the Results component of showing the first regression results, should be referenced in Lines 98 and 115 for the former and Lines 147 for the latter. 

We moved several topics discussed in the Results section to the Methods section. *****

• Line 60 (Introduction): I believe you meant ‘MERS-CoV’ here?

*****We greatly appreciate the reviewer catching our omission of the S in the abbreviation of Middle East Respiratory Syndrome (MERS). *****

• Lines 65-66 (Introduction): Perhaps you can expound a bit more on these evidence or suggestions, especially its quality? How about counter evidence or arguments? 

*****At the reviewers recommendation we added an entire summary of the existing peer-reviewed literature on prior investigations into the section to which we were referred. *****

• Line 79 (Methods): What do longitudinal changes mean? By time? Perhaps you can use a diﬀerent word to make this clearer.

*****We appreciate the reviewer alerting us to this ambiguous term. We substituted geographical for longitudinal. *****

• Lines 95-97 (Methods): Perhaps clearly mention that you are referring to ‘daily’ (new) conﬁrmed cases.

*****We added the word “daily” to assist in the clarification of the methods. *****

• Lines 100-102 (Methods): To be honest, I am more inclined to not discuss anything about longitude coverage given it does not aﬀect meteorological conditions, right? This is to save space or number of words.

*****While we agree that the most important aspect of the description is the discussion of temperature range, we also feel that the discussion of the global coverage offers the reader some reassurance that the 50 countries selected were not geographically biased. Moreover, we happen to be fortunate that the requirements of PLOS ONE is so accommodating: “Manuscripts can be any length. There are no restrictions on word count, number of figures, or amount of supporting information.” *****

• Lines 173-175 (Results): I think it would have been more appropriate (but admittedly, more complex) if the counts have been age-standardised with respect to a reference population? So no need for such covariate? Has this been explored/considered?

*****We chose to keep the Mean Age because it may have offered a hint at a cause for the age effect: the younger the mean age the greater the increase in rate of transmission. This is what we had hoped to suggest by stating that “social gathering behavior” might account for this effect. In the US the gathering of young individuals has been suggested to increase the rate of transmission, so this seemed like a plausible interpretation of the association. *****

• Table 2: I believe the 4th column refers to estimated regression coeﬃcients right? Perhaps just mention this instead of correlation as it might be a bit misleading?

*****In order to provide additional clarity as the reviewer suggested we change “The resulting regression coefficients…” to “The resulting estimated regression coefficients (which we labelled “Correlation” for simplicity)” *****

• Table 2: I am not too keen about reporting analyses with Recovered data in the same amount of details as with Conﬁrmed Cases and Deaths just because they may be of the poorest quality among the three, at least in many countries that I know of, Recoveries are almost not always reported. So this might give an impression that the data is also robust if included as part of the main analysis. May I suggest instead to remove this in the Table and just simply write out a paragraph for somewhere in the Results section.

*****While we completely understand the excellent point the reviewer is making, and do not trust recovered results nearly as much as we do confirmed and deaths cases, we chose to leave the recovered cases in the table for the following reasons:

1. The size (regression coefficient) and association (R^2) for recovered cases is greater than for deaths. 

2. We further propose this is the case because temperature impacts the virus and not the host. In other words, the survival and transmissibility of SARS-CoV-2 is directly affected by the temperature, whereas the likelihood that the infection will kill someone is relatively less influenced by the temperature. 

3. Since Recovered would be expected to more closely reflect the impact of temperature on viral transmissibility (with the more individuals in, the more individuals out) than deaths, we decided to let the data speak for itself, and allow the reader to draw their own conclusions. *****

• Table 2: In my experience, annotating p-values > 0.05 is not a standard wat of reporting. Often times you annotate those which are statistically signiﬁcant, not the other way around. Also, the use of scientiﬁc notation/exponentiation in p-values appear to obscure the values p-values (making them appear less in magnitude), so may I suggest changing to 3 or 4 decimal places and whenever less than 0.001 just say <0.001?

*****As recommended by the reviewer we removed the scientific notation. P values were converted to * according to this pattern: 0.05, 0.005, 0.0005, 0.00005, etc= *, **, ***, ****, etc*****

• Lines 236-237 (Results): Shouldn’t this be part of Methods?

*****We greatly appreciate the reviewers recognition that this information was indeed misplaced within our manuscript. It was moved to the end of the Method section as recommended. *****

• Lines 279-281 (Discussion): I think the way it is currently written it does not make sense. Longitudinal diﬀerences do not dictate atmospheric and meteorological conditions, so if the reference is to the model using temperature then it does not apply. But if the reference is to the approach but using diﬀerence covariates (say, cultural or population behaviour measures) then I see how it can also be useful. What I am trying to say is perhaps you can rephrase this better to avoid this ambiguity in interpretations.

*****We appreciate the reviewer’s continued improvement of our manuscript with her thoughtful, and detailed editorial criticisms, critique and recommendations. The use of the term “longitudinal” was confusing and we replaced the sentence with this: “whether our model is accurate for seasonal predictions, surveillance and preparedness related to the spread of the virus over different geographical areas.” *****

6. PLOS authors have the option to publish the peer review history of their article (what does this mean?). If published, this will include your full peer review and any akached ﬁles.

Do you want your identity to be public for this peer review? For information about this choice, including consent withdrawal, please see our Privacy Policy.

Reviewer #1: No

Reviewer #2: No

Reviewer #3: No

 [NOTE: If reviewer comments were as an attachment ﬁle, they will be attached to this email and accessible via the submission site. Please log into your account, locate the manuscript record, and check for the action link "View Attachments". If this link does not appear, there are no attachment ﬁles.ti

While revising your submission, please upload your ﬁgure ﬁles to the Preﬂight Analysis and Conversion Engine (PACE) digital diagnostic tool, hkps://pacev2.apexcovantage.com/. PACE helps ensure that ﬁgures meet PLOS requirements. To use PACE, you must ﬁrst register as a user. Registration is free. Then, login and navigate to the UPLOAD tab, where you will ﬁnd detailed instructions on how to use the tool. If you encounter any issues or have any questions when using PACE, please email PLOS at ﬁgures@plos.org. Please note that Supporting Information ﬁles do not need this step.

In compliance with data protection regulations, you may request that we remove your personal registration details at any time. (Remove my informa1on/details). Please contact the publication oﬃce if you have any questions.

*****Bibliography

1. Peter H Westfall RDT, Russell D Wolfinger. Multiple comparisons and multiple tests using SAS. Second Edition ed. Cary, NC USA: SAS Institute Inc.; 2011 July 2011. 391 p.

2. Holm S. A Simple Sequentially Rejective Multiple Test Procedure. Scandinavian Journal of Statistics. 1979;6(2):65-70.

3. Zhu L, Liu X, Huang H, Avellan-Llaguno RD, Lazo MML, Gaggero A, et al. Meteorological impact on the COVID-19 pandemic: A study across eight severely affected regions in South America. Sci Total Environ. 2020;744:140881.

4. Castillo RC, Staguhn ED, Weston-Farber E. The effect of state-level stay-at-home orders on COVID-19 infection rates. Am J Infect Control. 2020;48(8):958-60.

5. Kwok KO, Wong VWY, Wei WI, Wong SYS, Tang JW. Epidemiological characteristics of the first 53 laboratory-confirmed cases of COVID-19 epidemic in Hong Kong, 13 February 2020. Euro Surveill. 2020;25(16).

6. Krantz SG, Rao A. Level of underreporting including underdiagnosis before the first peak of COVID-19 in various countries: Preliminary retrospective results based on wavelets and deterministic modeling. Infect Control Hosp Epidemiol. 2020:1-3.

7. Yao Y, Pan JH, Liu ZX, Meng X, Wang WD, Kan HD, et al. No association of COVID-19 transmission with temperature or UV radiation in Chinese cities. Eur Respir J. 2020;55(5). *****

---

## [Decision Letter · Decision Letter 1]

6 Nov 2020

PONE-D-20-17772R1

Evidence and magnitude of the effects of meteorological changes on SARS-CoV-2 transmission: Penny wise, pandemic foolish?

PLOS ONE

Dear Dr. Kaplin,

Thank you for submitting your manuscript to PLOS ONE. After careful consideration, we feel that it has merit but does not fully meet PLOS ONE’s publication criteria as it currently stands. Therefore, we invite you to submit a revised version of the manuscript that addresses the points raised during the review process.

The Authors are expected to address all the criticisms by all Reviewers. In particular, the statement “We were therefore able to capture a period where weather and not social interventions played the predominant role in impacting transmission” may not be fully justified (Reviewers #3). In additional to the above comments, please address,

Title, please remove “Penny wise, pandemic foolish?” as it is not a conclusion or description of the studyL87-90, the exclusion of China, Russia and Italy and Southern Hemisphere countries is not well justified. The case numbers in different countries are available from different sources which were translated into English. Even Italy has a larger number of cases, the potential impact of meteorological variables should equally apply to the trend within country.How to assess consistency in reporting should be made explicit as inclusion or exclusion criteria.What was the reason to use data up to Apr 6? Variation in the meteorological variables is expected to be larger over summer and should better demonstrate the effect of interest.Table 1, Finland has a CT^-1 of 16.13 and Tmin = 19.3. However, this data point was missing in Figure 2. Also for Vietnam, the CT^-1 was 36.94 but the data point was missing in Figure 2. Please confirm accuracy of the data/analysis.Table 2, please clarify the label ‘correlation’. For LAPC, it was 21.244 when assessing confirmed cases which is out of usual range for correlation. Is it coefficient instead?Table 2, if the label ‘correlation’ is correct, please note that the correlations, even statistically significant, was mostly small (< 0.3), indicating a weak correlation.

We look forward to receiving your revised manuscript.

Kind regards,

Eric HY Lau, Ph.D.

Academic Editor

PLOS ONE

Additional Editor Comments (if provided):

The Authors are expected to address all the criticisms by all Reviewers. In particular, the statement “We were therefore able to capture a period where weather and not social interventions played the predominant role in impacting transmission” may not be fully justified (Reviewers #3). In additional to the above comments, please address,

1. Title, please remove “Penny wise, pandemic foolish?” as it is not a conclusion or description of the study

2. L87-90, the exclusion of China, Russia and Italy and Southern Hemisphere countries is not well justified. The case numbers in different countries are available from different sources which were translated into English. Even Italy has a larger number of cases, the potential impact of meteorological variables should equally apply to the trend within country.

3. How to assess consistency in reporting should be made explicit as inclusion or exclusion criteria.

4. What was the reason to use data up to Apr 6? Variation in the meteorological variables is expected to be larger over summer and should better demonstrate the effect of interest.

5. Table 1, Finland has a CT^-1 of 16.13 and Tmin = 19.3. However, this data point was missing in Figure 2. Also for Vietnam, the CT^-1 was 36.94 but the data point was missing in Figure 2. Please confirm accuracy of the data/analysis.

6. Table 2, please clarify the label ‘correlation’. For LAPC, it was 21.244 when assessing confirmed cases which is out of usual range for correlation. Is it coefficient instead?

7. Table 2, if the label ‘correlation’ is correct, please note that the correlations, even statistically significant, was mostly small (< 0.3), indicating a weak correlation.

Reviewers' comments:

Reviewer's Responses to Questions

**Comments to the Author**

1. If the authors have adequately addressed your comments raised in a previous round of review and you feel that this manuscript is now acceptable for publication, you may indicate that here to bypass the “Comments to the Author” section, enter your conflict of interest statement in the “Confidential to Editor” section, and submit your "Accept" recommendation.

Reviewer #1: All comments have been addressed

Reviewer #2: All comments have been addressed

Reviewer #3: All comments have been addressed

2. Is the manuscript technically sound, and do the data support the conclusions?

Reviewer #1: Partly

Reviewer #2: (No Response)

Reviewer #3: Yes

3. Has the statistical analysis been performed appropriately and rigorously? 

Reviewer #1: I Don't Know

Reviewer #2: (No Response)

Reviewer #3: Yes

4. Have the authors made all data underlying the findings in their manuscript fully available?

Reviewer #1: Yes

Reviewer #2: (No Response)

Reviewer #3: Yes

5. Is the manuscript presented in an intelligible fashion and written in standard English?

Reviewer #1: Yes

Reviewer #2: (No Response)

Reviewer #3: Yes

6. Review Comments to the Author

Reviewer #1: The manuscript seems to be clearer with improved grammar. The figures seem to be correctly labelled.

Reviewer #2: (No Response)

Reviewer #3: I commend the authors for a much improved version of the manuscript. It is now at a more readable state without compromising nor overselling the strength of their findings. While I still personally think that it was not fully justified that Southern Hemisphere countries should be excluded, as I think they can still be considered even as a sub-analysis of its own or just part of the sensitivity analysis which if things went as expected should be able to further strengthen their claims, I still think that my personal opinion should not hinder the publication of such a very timely and still informative article. Indeed, their key findings along of potential seasonality in pathogen transmission of SARS-CoV-2 are intriguing and very useful enough to be published, as this should warrant further investigation of plausible biological underpinnings of these findings.

I do have some minor comments that I hope the authors consider in preparing the final version of this piece:

1.) In the Introduction, lines 80-95, while I understand that the cited literature presented findings in units of Celsius for temperature, the authors can consider, even written in parentheses, converted units to Fahrenheit for consistency in the use of temp units all throughout the study, w/c may also improve comparability of their findings with others'.

2.) At first, I thought that their inclusion criteria were based on daily new case counts (hence, the reason for the way I stated my question previously w/c also confused the authors), but it was made clear to me that they were actually working with cumulative case counts. If that is the case, I think dropping the term 'daily' should already suffice as the explanation is now clear as long as the term 'cumulative' is retained.

3.) In Line 175, what does '/' mean in CT/CT^(-1)? I trust that this is read as 'or' rather than the mathematical symbol for division (as it will just turn out to be CT^2 w/c doe snot makes sense nor even appeared anywhere in the manuscript). Am I right about this? If so, perhaps change '/' to spell out 'or' in full to avoid confusion.

4.) To my mind, the last two subsections in the Results section (relationship of CT to Td, and model equation formulation) fit better as last subsections of the Methods section as they set the stage on as how to interpret their results moving forward. Can the authors consider moving these?

5.) Lastly, in lines 367-368, I still find the statement "We were therefore able to capture a period where weather and not social interventions played the predominant role in impacting transmission." a very strong statement considering the authors considered a long time frame, and there are other regions/countries who almost immediately, and their citizens' abided by, imposed rules on social gathering restrictions and physical distancing. In other words, this claim is untestable and is far too stretched, perhaps my suggestion is to 'soften' the claim by using more nuanced words/qualifiers such as "where weather MAY have played more role in impacting transmission as evidenced by ..."

These are all I have. Again, thank you for the opportunity to review this well-written piece.

7. PLOS authors have the option to publish the peer review history of their article (what does this mean?). If published, this will include your full peer review and any attached files.

Reviewer #1: No

Reviewer #2: No

Reviewer #3: No

---

## [Author Response · Author response to Decision Letter 1]

1 Dec 2020

Dear PLOS ONE Journal Editor & Reviewers,

It has been 5.7 months since our initial submission to PLOS ONE and as the current exponential increase in COVID-19 demonstrates we were absolutely correct about the temperature sensitivity of SARS-CoV-2 and our findings foretold the current global situation. The chance to anticipate the exponential rise in COVID-19 transmission has passed, but there is still a final opportunity to clarify emphatically and without ambiguity the following: that the magnitude of the effect of decreasing temperature on the rise in COVID-19 transmission that we predicted in our manuscript was spot on. Through publishing our manuscript in PLOS ONE now it can be hoped that there is still the opportunity to shed light on the full impact of the cold temperature and anticipate the improvement with the warm temperature that will return in the spring.

We have answered all of the questions and addressed all of the comments below, not by using mere conjecture or opinion, but by providing evidence for all of the explanations we provide—three examples are worth underscoring. Frist, we obtained daily facial covering policies (available for 45 out of the 50 countries in our analysis) to demonstrate that non-pharmaceutical interventions did not start in earnest until after 4/6/20—thereby supporting with evidence our assertion that we were able to “capture a period where weather and not social interventions played the predominant role in impacting transmission.” Second, we also did a complete analysis of the daily rate of SARS-CoV-2 testing in each of the 50 countries we analyzed, as a surrogate marker of consistency in reporting COVID-19 cases, and then demonstrated through adding this variable to our multiple regression model that it did not change our findings. Third, we did an exhaustive search of the literature to demonstrate that studies investigating the impact of temperature on COVID-19 transmission that included Southern Hemisphere countries, but otherwise used a similar approach to ours, resulted in either no effect of temperature or an effect that contradicts all prior findings of seasonality in coronaviruses. This is in contrast to studies like ours that used only Northern Hemisphere countries in their analysis, which were able to resolve a clear effect supportive of seasonality in SARS-CoV-2. We have speculated previously why the combination of Northern and Southern Hemispheres countries obscures the signal from each hemisphere. Perhaps it is—as we previously speculated--that in the Northern Hemisphere the combined influences of spread from person to person (increasing transmission) and the warming temperature (decreasing transmission) is not symmetrical with the Southern Hemisphere with the contributions of the person to person spread (increasing transmission) and cooling temperature (increasing transmission). But regardless of the reason, the findings from published papers to date argue that analysis of the Northern Hemisphere independent of the Southern Hemisphere during the early phases of the COVID-19 pandemic is necessary to tease out the influence of temperature on viral transmission. 

We believe that the manuscript has been greatly improved by many of the comments and recommendations of the editors and reviewers. We hope that with our current responses we will be able to move forward to publishing this work while it can still have an impact on current discussions of the seasonality of SARS-CoV-2 transmission. The fact that our predictions are being borne out as we submit this second response rebuttal is providing empirical validation of our theoretical work. 

Please note that our responses below are separated by strings of “*” both before and after. Also note that there are two sections of comments, separated by the questions provided in the PLOS ONE decision letter, where we provide responses: lines 65-300, and lines 396-455.

Subject: PLOS ONE Decision: Revision required [PONE-D-20-17772R1] - [EMID:ef2d59aa1310493b]

Date: Friday, November 6, 2020 at 5:22:02 AM Eastern Standard Time

From: em.pone.0.6f2248.ce0984c5@editorialmanager.com on behalf of PLOS ONE

To: Adam Ian Kaplin

PONE-D-20-17772R1

Evidence and magnitude of the eﬀects of meteorological changes on SARS-CoV-2 transmission: Penny wise, pandemic foolish?

PLOS ONE

Dear Dr. Kaplin,

Thank you for submitting your manuscript to PLOS ONE. A]er careful consideration, we feel that it has merit but does not fully meet PLOS ONE’s publication criteria as it currently stands. Therefore, we invite you to submit a revised version of the manuscript that addresses the points raised during the review process.

The Authors are expected to address all the criticisms by all Reviewers. In particular, the statement “We were therefore able to capture a period where weather and not social interventions played the predominant role in impacting transmission” may not be fully justiﬁed (Reviewers #3): “I still ﬁnd the statement "We were therefore able to capture a period where weather and not social interventions played the predominant role in impacting transmission." a very strong statement considering the authors considered a long time frame, and there are other regions/countries who almost immediately, and their citizens' abided by, imposed rules on social gathering restrictions and physical distancing. In other words, this claim is untestable and is far too stretched, perhaps my suggestion is to 'soften' the claim by using more nuanced words/qualiﬁers such as "where weather MAY have played more role in impacting transmission as evidenced by ..."

• We appreciate the opportunity to explain why we maintain that we captured a period where weather and not social interventions played the predominant role in impacting transmission, especially in light of the fact that we believe that this is one of the important reasons we found such a robust and quantifiable impact of temperature on SARS-CoV-2 whereas others did not. 

• To demonstrate the timing of our sampling (i.e. 1/22/20-4/6/20) relative to non-pharmaceutical interventions, we consulted the University of Oxford database that has tracked global government responses to COVID-19:

o The details of the Oxford database can be found here: Variation in government responses to COVID-19, October 2020, Blavatnik School of Government, University of Oxford, https://www.bsg.ox.ac.uk/research/research-projects/coronavirus-government-response-tracker

o As an example of non-pharmaceutical interventions implemented by countries we included in our analysis, we evaluated the timing of policies on the use of facial coverings outside the home. Forty-five out of the 50 countries we analyzed had available date in the Oxford database for facial covering policy. This is an intervention that has been recently studied, that with respect to masks the authors concluded that “rational support for this strategy is stronger than, for instance, that for recommendations on disinfection of inanimate surfaces” (1). See attached response to reviewers figure 1 titled FACIAL COVERING BY COUNTRY, where the rating scale is as follows: 0- No policy, 1- Recommended, 2- Required in some specified shared/public spaces outside the home with other people present, or some situations when social distancing not possible, 3- Required in all shared/public spaces outside the home with other people present or all situations when social distancing not possible, 4- Required outside the home at all times regardless of location or presence of other people.

o In the attached figure titled “FACIAL COVERING BY COUNTRY,” the top of four sections represents the use of facial coverings by the 45 countries analyzed in the manuscript during the dates 1/22/20-4/6/20. Grey indicates a score of 0=No policy. The majority of countries (76%) did not institute any facial covering policies (even recommendations) by 4/6/20, and as late as one week prior to the end of the sampling period 84% of countries had no policy. By contrast, one month after the end of our sampling period (i.e. after 4/6/20) 58% had a facial covering policy, and by two months after the end of our sampling period 75% of countries had instituted facial covering policies. 

o Thus, the 45 of the 50 countries that we analyzed over the relevant timeframe (1/22-4/6/20) had minimal to no non-pharmaceutical interventions, lending strong evidence to the assertion that we “captured a period where weather and not social interventions played the predominant role in impacting transmission.”

In additional to the above comments, please address,

1. Title, please remove “Penny wise, pandemic foolish?” as it is not a conclusion or description of the study

“Penny wise, pandemic foolish?” was removed from title.

2. L87-90, the exclusion of China, Russia and Italy and Southern Hemisphere countries is not well justiﬁed. The case numbers in diﬀerent countries are available from diﬀerent sources which were translated into English. Even Italy has a larger number of cases, the potential impact of meteorological variables should equally apply to the trend within country.

Exclusion of China:

• We excluded countries with multiple data sites (for both COVID-19 cases and NOAA meteorological data) within its borders. Each country had differing responses to the increasing rates of COVID-19 within its borders (e.g. the timing and availability of testing, government policy decisions about when and how robustly to respond with non-pharmaceutical interventions, etc). Because each country that was included contributed data representing a single response to COVID-19, the variability in the response would be randomized (some countries more and some less in various aspects of their response). The inclusion of a single country with numerous measures within its borders would risk creating a systematic bias for that country in terms of a potential effect on the rate of transmission of COVID-19. China had 33 provinces for which cases were collected from JHU COVID-19 Data Repository by the Center for Systems Science and Engineering (CSSE) at Johns Hopkins University (https://github.com/CSSEGISandData/COVID-19). 

• If one chooses just the single province in China from which the majority of cases occurred, Hubei, there is another reason we did not include this in our analysis: late stage sampling. In the interest of conformity of data source utilization, we used the JHU COVID-19 Data Repository database that begins on 1/22/20. As can be seen from the attached figure titled “Rebuttal Hubei China Confirmed Cases” the beginning of data collection starts late in the course of the rise of cumulative confirmed cases—i.e. 54% of the plateau of the log of cumulative confirmed cases had been achieved in Hubei by 1/22/20. Thus, it is not possible to capture the early initial rate of increases that, as explained above on lines 28-60, is key to our approach in that it allows us to isolate the SARS-CoV-2 transmission period that is predominantly influenced by temperature prior to non-pharmaceutical interventions were employed. 

• There is also abundant evidence to support the dramatic under-reporting of cases from China before the time the JHU COVID-19 Data Repository database begins on 1/22/20, thus suggesting that trying to base results on non-JHU sources would be fraught with large inaccuracies. For example, a recent publication by Krantz et al (2) calculated the ratio of reported COVID-19 cases and contrasted them with model-based predictions of COVID-19 for 8 major countries. The ratio of reported to infected cases for China was between 1:149 to 1:1104. In comparison, France was 1:5 and Germany was 1:4. Based on cremation related information from China, He et al (3) were able to calculate COVID-19 case estimates projected on February 7, 2020 in Wuhan ranged from 305,000 to 1,272,000 for infections that were at least 10 times the official figures of 13,603. The same source of information implied starting time of the outbreak is October 2019, rather than the government reported date of 12/31/19 (3).

Exclusion of Russia:

• Russia presented the problem that a country of its magnitude had only a single JHU CSSE COVID-19 report of cases with a single latitude and longitude, which could clearly not accurately represent the appropriate temperature corresponding to the rate of increase of cases for the entire country. As we suggested in our manuscript, one of the reasons for our robust findings was that “we took great care to ensure that we used weather stations located as close to the longitude and latitude corresponding to the cases in each country as presented by the Johns Hopkins real time COVID-19 tracking database.” Russia has the largest land area of any country in the world, with vastly differing temperatures across its vast expanse, being comprised of 17 million square kilometers. The largest country included in the 50 analyzed in our manuscript was Algeria, that has a land area of 2.4 million square kilometers, which is 7-fold smaller than Russia. Obviously one temperature for such a vast country with a huge range of local climates could not possibly yield accurate data. 

• There are also reports from the media that the Russian government had similar problems with under-reporting, but details from the published literature were not available. 

• Not surprisingly, we included Russia in the analysis with the other 50 countries we originally analyzed and obtained a calculated R2 (0.37) and the p value (1.42E-06) of the regression of CT-1 vs Tmin_Median. The R2 was 22% smaller and the p-value 23-fold larger than without including Russia in the analysis, with Russia being a clear outlier in the analysis. This demonstrates how Russia’s deviation from the anticipated pattern is precisely what would be predicted from the above considerations.

Exclusion of Italy:

• We excluded Italy because it suffered from a vast degree of under-testing resulting in a dramatic under-reporting of cases of COVID-19. Lau et al (4) used the mortality rate as the main indicator to evaluate the extent of underreporting and under-detection of COVID-19 cases in various countries. Using this approach the authors concluded that there were an extremely high number of undetected COVID-19 cases in Italy, with less than 2 percent of COVID-19 cases being subjected to testing and consequently reported (4). 

• To investigate if including Italy in the analysis of the relationship between CT-1 and Tmin_Median of the original 50 country cohort, we reran the regression using the total of 51 countries by including Italy. Interestingly the R2 and p value were essentially unchanged. This suggests that the under-reporting of cases due to under-testing (as described above) was likely systematic and continued throughout the period of ascertainment of 1/22-4/6/20. Nevertheless, given the known deviation in Italy from the reporting in the other 50 countries of COVID-19 prevalence—with only 2% of cases being accounted for in Italy—we decided to exclude Italy because, even if the slope of increasing cumulative cases of transmission was correct, the initial regression of the log of cumulative confirmed cases vs time was misleading in magnitude.

Exclusion of Southern Hemisphere: 

1. We were concerned about the possible asymmetry of the effects of changing temperatures on the increasing vs decreasing rates of transmission as described below. Here is why we believe our approach to analyzing the association of the rate transmission with temperature was best limited to the northern hemisphere:

a. See lines 19-31 above for additional considerations of the asymmetry of data from Northern vs Southern Hemisphere countries.

b. Both the origin of transmission in the northern hemisphere, and the rates of global spread of COVDI-19 faster in the northern latitudes than the southern, meant that the data available for southern hemisphere countries was significantly restricted over the time period we sampled (especially when one considered the trailing rates of deaths). 

c. Sampling from a later time from Southern latitudes would be further complicated by the fact that lessons learned by observing the spread in Northern neighbors could have exerted a different response in the Southern countries, thereby affecting early rates of transmission. 

2. The most compelling reason not to include Southern Hemisphere with Northern Hemisphere countries is that studies that have done so have resulted in negative or contradictory findings, including one published by PLOS ONE on 10/22/20, whereas studies that evaluated predominantly (i.e. 99.8% of cases) Northern Hemisphere countries found a clear impact of temperature on COVID-19 transmission.

a. A study by Islam et al examined COVID-19 cases (during 1/22 - 4/20/20) from both Northern and Southern Hemispheres from the database developed by Johns Hopkins University (5). They found a positive association between temperature and COVID-19 cases, contradicting the normal findings of seasonality in respiratory viruses where there is an inverse correlation between temperature and viral transmission. They concluded that their findings “may suggest that warm environment may provide a more favorable condition for SARS-CoV-2 survival, which is conflicting with the pattern proposed for other coronavirus such as SARS-CoV and MERS-CoV” (5). Thus, using the same database that we employed and only a slightly longer timeframe, but employing Southern Hemisphere countries while we only Northern, they got opposite results. 

b. A study by Kumar et al published in PLOS ONE examined COVID-19 cases between 1/22-4/3/20, much like our study that investigated between 1/22-4/6/20 (6). But instead of taking 50 representative countries from the Northern Hemisphere as we did, the authors chose 67 countries from both the Northern and Southern Hemispheres. Unlike our highly significant findings, where adjusted R2=0.610 and p= 1.45E-06, Kumar et al “did not observe the statistically significant association of mean temperature and spread of the COVID-19” (6).

c. In contrast, a study published in JAMA Network that investigated 99.8% of cases drawn from northern hemisphere countries found a statistically significant finding: “The 8 cities with substantial community spread as of March 10, 2020, were located on a narrow band, roughly on the 30 – 50 N corridor” (7). 

Thus, the best evidence for our assertion that including Southern Hemisphere in the analysis would conflict with the signal we obtained from Northern Hemisphere countries is borne out by the negative findings of studies that employed the former approach and the success of the use of the latter. 

3. Reviewer #3 who originally raised the issue of the Sothern Hemisphere, and was the only reviewer to even bring up the topic, concluded in her evaluation of our second submission, concluded with respect to omission of the Southern Hemisphere that “I still think that my personal opinion should not hinder the publication of such a very timely and still informative article.” 

3. How to assess consistency in reporting should be made explicit as inclusion or exclusion criteria.

• We maintain that the variation in consistency in reporting by each country should be addressed by the inclusion of 50 countries, based on specific criteria, where the differences should cancel each other out, and note that we could find no other published article addressing seasonality of COVID-19 or related viruses for which consistency in reporting was specifically addressed for this reason. See discussion above in lines 115-124 for further details.

• However, to provide additional support for the lack of contribution of consistency in reporting on our specific findings we invested considerable effort in demonstrating that fact by utilizing a surrogate measure of reporting consistency--SARS-CoV-2 testing over time in countries included in our analysis. Since the reporting of cases is dependent on, and proportional to, the extent of viral testing that was performed, including an independent variable in our multiple regression for SARS-CoV-2 testing per capita in each country over the relevant time period (i.e. 1/22-4/6/20) should provide a means to address the question of the influence of reporting consistency. 

o SARS-CoV-2 testing is tracked by FIND. FIND is the Foundation for Innovative New Diagnostics, which is a global non-profit organization driving innovation in the development and delivery of diagnostics to combat major diseases FIND is a WHO Collaborating Centre for Laboratory Strengthening and Diagnostic Technology Evaluation. https://www.finddx.org/covid-19/test-tracker/ : “Testing coverage is a critical component of pandemic response, providing a view on the data available to inform policy and monitor the effectiveness of public health measures. While many countries do not publish official numbers of tests conducted, others are doing this across individual websites, statistical reports and press releases, often in multiple languages and updated with different periodicity. FIND is working to build a global picture of the testing coverage for COVID-19 and facilitate the accurate interpretation and study of case and death numbers.”

o Daily SARS-CoV-2 testing was available for 45/50 countries included in our analysis—only Iraq, Kuwait, Lebanon, Moldova, Oman did not have available data. We took the slope of the log of the cumulative testing per capita for each country, and included it in our multivariate regression to establish the correlation between median minimal temperature and rate of increase of the log of confirmed cases.

o Including the slope of the log of SARS-CoV-2 testing for each country in the regression of the Coefficient of Time (i.e. CT vs Tmin_Median), or the inverse slope of the log of SARS-CoV-2 testing in the regression of the inverse Coefficient of Time (CT-1 vs Tmin_Median) did not appreciably affect the outcome of the analysis (e.g. the R2 was changed from 0.30 to 0.37 for the former, and 0.45 to 0.46 for the latter). 

o Thus, utilizing the daily amount of SARS-CoV-2 testing in each country as a surrogate measure of their consistency in reporting did not change the outcome of our analysis. 

4. What was the reason to use data up to Apr 6? Variation in the meteorological variables is expected to be larger over summer and should better demonstrate the eﬀect of interest.

• See explanation above (lines 28-61) to the concern about our justification of the assertion that: “We were therefore able to capture a period where weather and not social interventions played the predominant role in impacting transmission.”

o As demonstrated, the period from 1/22-4/6/20 was specifically selected to capture the period of time when the increase in the rate of transmission of COVID-19 was driven predominantly by meteorological variables such as temperature and before non-pharmaceutical interventions were instituted. 

o We demonstrated this assertion by showing that between 1/22-4/6/20 the institution of recommendations for facial covering was largely not started in most of the countries examined. See above lines 67-100 for detailed discussion. 

5. Table 1, Finland has a CTti-1 of 16.13 and Tmin = 19.3. However, this data point was missing in Figure 2. Also for Vietnam, the CTti-1 was 36.94 but the data point was missing in Figure 2. Please conﬁrm accuracy of the data/analysis.

• The graph is correct as labelled—it shows the predicted CT-1 vs Tmin_Med. 

• To clarify this further in the figure legend we made the following changes (designated as italicized and in red): 

o “Figure 2: shows for fifty representative countries the predicted regression of the reciprocal of the rate of increase of the log of confirmed cases (CT-1) vs the median minimal temperature. Predicted values (i.e. fitted values) are the values that the model predicts using the regression equation generated from the multiple regression. This establishes the relationship between temperature and the rate of daily increase of confirmed cases of COVID-19. The multiple regression of CT-1 for all countries yielded a significant correlation (adjusted R2=0.610, p=1.45 x 10-6) using covariates Tmin_Med, median age, land area per capita and days of cases.

• See also the included graph titled “Rebuttal FIGURE 2 with & without Predicted Values” that demonstrates the difference between the plot with and without using predicted values. 

6. Table 2, please clarify the label ‘correlation’. For LAPC, it was 21.244 when assessing conﬁrmed cases which is out of usual range for correlation. Is it coeﬃcient instead?

• This was raised in the first review (see lines 930-935 in our original rebuttal letter) and we attempted to clarify by including in the Table legend the following: “Shown are the resulting estimated regression coefficients (which we labelled “Correlation” for simplicity) and the R2 for each regression.”

• To make this even less confusing, we substituted the term “Regression Coefficients” for “Correlation”, and changed the corresponding sentence in the Table legend to read: “Shown are the resulting estimated regression coefficients and the R2 for each regression.”

7. Table 2, if the label ‘correlation’ is correct, please note that the correlations, even statistically signiﬁcant, was mostly small (< 0.3), indicating a weak correlation. Please submit your revised manuscript by Dec 21 2020 11:59PM. If you will need more time than this to complete your revisions, please reply to this message or contact the journal oﬃce at plosone@plos.org. When you're ready to submit your revision, log on to https://www.editorialmanager.com/pone/ and select the 'Submissions Needing Revision' folder to locate your manuscript ﬁle.

• As indicated above, the label has been changed to Regression Coefficients for clarification. 

• As shown in the Adjusted R2 column, our correlation was large, reaching a maximum of 0.610.

A rebuttal letter that responds to each point raised by the academic editor and reviewer(s). You should upload this letter as a separate ﬁle labeled 'Response to Reviewers'.

A marked-up copy of your manuscript that highlights changes made to the original version. You should upload this as a separate ﬁle labeled 'Revised Manuscript with Track Changes'.

An unmarked version of your revised paper without tracked changes. You should upload this as a separate ﬁle labeled 'Manuscript'.

If you would like to make changes to your ﬁnancial disclosure, please include your updated statement in your cover letter. Guidelines for resubmittng your ﬁgure ﬁles are available below the reviewer comments at the end of this letter.

If applicable, we recommend that you deposit your laboratory protocols in protocols.io to enhance the reproducibility of your results. Protocols.io assigns your protocol its own identiﬁer (DOI) so that it can be cited independently in the future. For instructions see: http://journals.plos.org/plosone/s/submission-guidelines#loc- laboratory-protocols

We look forward to receiving your revised manuscript. Kind regards,

Eric HY Lau, Ph.D. Academic Editor PLOS ONE

Reviewers' comments:

Reviewer's Responses to Questions

Comments to the Author

1. If the authors have adequately addressed your comments raised in a previous round of review and you feel that this manuscript is now acceptable for publication, you may indicate that here to bypass the “Comments to the Author” section, enter your conﬂict of interest statement in the “Conﬁdential to Editor” section, and submit your "Accept" recommendation.

Reviewer #1: All comments have been addressed

Reviewer #2: All comments have been addressed Reviewer #3: All comments have been addressed

2. Is the manuscript technically sound, and do the data support the conclusions?

The manuscript must describe a technically sound piece of scientiﬁc research with data that supports the conclusions. Experiments must have been conducted rigorously, with appropriate controls, replication, and sample sizes. The conclusions must be drawn appropriately based on the data presented.

Reviewer #1: Partly Reviewer #2: (No Response) Reviewer #3: Yes

3. Has the statistical analysis been performed appropriately and rigorously? Reviewer #1: I Don't Know

Reviewer #2: (No Response) Reviewer #3: Yes

4. Have the authors made all data underlying the ﬁndings in their manuscript fully available?

The PLOS Data policy requires authors to make all data underlying the ﬁndings described in their manuscript fully available without restriction, with rare exception (please refer to the Data Availability Statement in the manuscript PDF ﬁle). The data should be provided as part of the manuscript or its supporting information, or deposited to a public repository. For example, in addition to summary statistics, the data points behind means, medians and variance measures should be available. If there are restrictions on publicly sharing data—e.g. participant privacy or use of data from a third party—those must be speciﬁed.

Reviewer #1: Yes

Reviewer #2: (No Response) 

Reviewer #3: Yes

5. Is the manuscript presented in an intelligible fashion and written in standard English?

PLOS ONE does not copyedit accepted manuscripts, so the language in submitted articles must be clear, correct, and unambiguous. Any typographical or grammatical errors should be corrected at revision, so please note any speciﬁc errors here.

Reviewer #1: Yes

Reviewer #2: (No Response) 

Reviewer #3: Yes

6. Review Comments to the Author

Reviewer #1: The manuscript seems to be clearer with improved grammar. The ﬁgures seem to be correctly labelled. Reviewer #2: (No Response)

Reviewer #3: I commend the authors for a much improved version of the manuscript. It is now at a more readable state without compromising nor overselling the strength of their ﬁndings. While I still personally think that it was not

 fully justiﬁed that Southern Hemisphere countries should be excluded, as I think they can still be considered even as a

 sub-analysis of its own or just part of the sensitivity analysis which if things went as expected should be able to

 further strengthen their claims, I still think that my personal opinion should not hinder the publication of such a very

 timely and still informative article. Indeed, their key ﬁndings along of potential seasonality in pathogen transmission

 of SARS-CoV-2 are intriguing and very useful enough to be published, as this should warrant further investigation of

 plausible biological underpinnings of these ﬁndings.

I do have some minor comments that I hope the authors consider in preparing the ﬁnal version of this piece:

1.) In the Introduction, lines 80-95, while I understand that the cited literature presented ﬁndings in units of Celsius for temperature, the authors can consider, even written in parentheses, converted units to Fahrenheit for consistency in the use of temp units all throughout the study, w/c may also improve comparability of their ﬁndings with others'. 

The conversion was made as advised. 

2.) At ﬁrst, I thought that their inclusion criteria were based on daily new case counts (hence, the reason for the way I stated my question previously w/c also confused the authors), but it was made clear to me that they were actually working with cumulative case counts. If that is the case, I think dropping the term 'daily' should already suﬃce as the

 explanation is now clear as long as the term 'cumulative' is retained.

The term “daily” was dropped and “cumulative” retained.

2.) In Line 175, what does '/' mean in CT/CTti(-1)? I trust that this is read as 'or' rather than the mathematical symbol for division (as it will just turn out to be CTti2 w/c doe snot makes sense nor even appeared anywhere in the manuscript). Am I right about this? If so, perhaps change '/' to spell out 'or' in full to avoid confusion.

All instances of “CT/CT-1” were changed to “CT or CT-1”.

3.) To my mind, the last two subsections in the Results section (relationship of CT to Td, and model equation formulation) ﬁt better as last subsections of the Methods section as they set the stage on as how to interpret their results moving forward. Can the authors consider moving these?

Relationship of CT to Doubling Time (Td)

• We incorporated the methods used to calculate the Td and its relationship to CT into the methods section, while keeping the actual results of the methodology in the Results section. 

• We note that to understand the methodology employed to explore the CT relationship with Td we needed to reference the results section. This was necessary because the introduction to the calculation of CT did not begin until the section titled Analysis of the impact of changing temperature on the rate of rise of COVID-19 cases in the results section.

• We hope that the reviewer prefers the inclusion of the methodology of the Td as it relates to CT in the methodology section with reference to the results section to fully understand the discussion, rather than leaving the entire discussion of the Td topic until after the relevant results concepts were covered. 

Details of the model formulations (including equations in appropriate context).

• We left the details of the model formulation at the end of the results section because when we tried moving it to the Methods section we found it more confusing than leaving it where it was already located. 

• The reason for the confusion when moved is that the equations detailed in the Model Formulations are not explained until the results section, where they are systematically detailed in a sequential manner to permit the reader to follow the logic employed in our analysis. The level of detail necessary to try and explain, out of context, the derivation of the models employed in the analysis we felt would lead to unnecessary duplication with the additional risk of having the reader not understand the step-by-step logic employed.

• We therefore decided to leave the details of model formulation to the end of the Results section, where it serves to summarize the highlights of the of the results section.

4.) Lastly, in lines 367-368, I still ﬁnd the statement "We were therefore able to capture a period where weather and not social interventions played the predominant role in impacting transmission." a very strong statement considering the authors considered a long time frame, and there are other regions/countries who almost immediately, and their citizens' abided by, imposed rules on social gathering restrictions and physical distancing. In other words, this claim is untestable and is far too stretched, perhaps my suggestion is to 'soften' the claim by using more nuanced words/qualiﬁers such as "where weather MAY have played more role in impacting transmission as evidenced by ..."

• Please see comments in lines 67-100 above, in response to the editors flagging this topic as the first issue they requested we explain. We present evidence from an University of Oxford database, which tracks daily government responses to COVID-19, that demonstrates the facial covering policy in 45 of the 50 countries that had available data did not substantially begin until after the 1/22-4/6/20 period we studied. We provide evidence above, using facial covering policies for the countries we analyzed as a marker of social intervention, that the timeframe we studied captured a period where weather and not social interventions played the predominant role in impacting transmission. 

These are all I have. Again, thank you for the opportunity to review this well-written piece.

7. PLOS authors have the option to publish the peer review history of their article (what does this mean?). If published, this will include your full peer review and any attached ﬁles.

Do you want your identity to be public for this peer review? For information about this choice, including consent withdrawal, please see our Privacy Policy.

Reviewer #1: No

Reviewer #2: No

Reviewer #3: No

[NOTE: If reviewer comments were submitted as an attachment ﬁle, they will be attached to this email and accessible via the submission site. Please log into your account, locate the manuscript record, and check for the action link "View Attachments". If this link does not appear, there are no attachment ﬁles.]

While revising your submission, please upload your ﬁgure ﬁles to the Preﬂight Analysis and Conversion Engine (PACE) digital diagnostic tool, https://pacev2.apexcovantage.com/. PACE helps ensure that ﬁgures meet PLOS requirements. To use PACE, you must ﬁrst register as a user. Registration is free. Then, login and navigate to the UPLOAD tab, where you will ﬁnd detailed instructions on how to use the tool. If you encounter any issues or have any questions when using PACE, please email PLOS at ﬁgures@plos.org. Please note that Supporting Information ﬁles do not need this step.

In compliance with data protec1on regula1ons, you may request that we remove your personal registra1on details at any 1me. (Remove my informa1on/details). Please contact the publica1on oﬃce if you have any ques1ons.

REFERENCES:

1. Fortaleza CR, Souza LDR, Rugolo JM, Fortaleza C. COVID-19: What we talk about when we talk about masks. Rev Soc Bras Med Trop. 2020;53:e20200527.

2. Krantz SG, Rao A. Level of underreporting including underdiagnosis before the first peak of COVID-19 in various countries: Preliminary retrospective results based on wavelets and deterministic modeling. Infect Control Hosp Epidemiol. 2020:1-3.

3. Mai He MD, Ph.D., 1 Li Li, M.A.2, Louis P. Dehner, M.D.,1 Lucia F. Dunn, Ph.D.3. Cremation based estimates suggest significant under- and delayed reporting of COVID-19

epidemic data in Wuhan and China. medRxiv. 2020.

4. Lau H, Khosrawipour T, Kocbach P, Ichii H, Bania J, Khosrawipour V. Evaluating the massive underreporting and undertesting of COVID-19 cases in multiple global epicenters. Pulmonology. 2020.

5. Islam N, Bukhari Q, Jameel Y, Shabnam S, Erzurumluoglu AM, Siddique MA, et al. COVID-19 and climatic factors: A global analysis. Environ Res. 2020:110355.

6. Kumar A, Misra S, Verma V, Vishwakarma RK, Kamal VK, Nath M, et al. Global impact of environmental temperature and BCG vaccination coverage on the transmissibility and fatality rate of COVID-19. PLoS One. 2020;15(10):e0240710.

7. Sajadi MM, Habibzadeh P, Vintzileos A, Shokouhi S, Miralles-Wilhelm F, Amoroso A. Temperature, Humidity, and Latitude Analysis to Estimate Potential Spread and Seasonality of Coronavirus Disease 2019 (COVID-19). JAMA Netw Open. 2020;3(6):e2011834.

---

## [Editor Report · Decision Letter 2]

4 Jan 2021

PONE-D-20-17772R2

Evidence and magnitude of the effects of meteorological changes on SARS-CoV-2 transmission

PLOS ONE

Dear Dr. Kaplin,

Thank you for submitting your manuscript to PLOS ONE. After careful consideration, we feel that it has merit but does not fully meet PLOS ONE’s publication criteria as it currently stands. Therefore, we invite you to submit a revised version of the manuscript that addresses the points raised during the review process.

The Authors have clarified most of the concern. However, it would be helpful to also include these explanations especially on the exclusion criteria in the manuscript, considering the following:

In the response, the authors provided valid reasons for the exclusion of China and Russia, especially the large variation in latitude and hence meteorological parameters. In the manuscript, please state this explicitly as an exclusion criterion. Under-reporting of infections is likely common but the extent can vary greatly across countries. However, this would only affect the results if the level of under-reporting changed significantly over time within country. Please also note that Krantz et al and Lau et al. only gave consistent conclusion on under-reporting for the US and Spain, but very different for China (least under-reporting in Lau et al.), Italy and France. In any case, please provide evidence in the manuscript on the under-reporting / change in under-reporting over time if this is still the main reason for exclusion.Please reconsider if language is really a criterion for exclusion. I believe different languages were used in the included countries.

We look forward to receiving your revised manuscript.

Kind regards,

Eric HY Lau, Ph.D.

Academic Editor

PLOS ONE

Additional Editor Comments (if provided):

The Authors have clarified most of the concern. However, it would be helpful to also include these explanations especially on the exclusion criteria in the manuscript, considering the following:

1. In the response, the authors provided valid reasons for the exclusion of China and Russia, especially the large variation in latitude and hence meteorological parameters. In the manuscript, please state this explicitly as an exclusion criterion. Under-reporting of infections is likely common but the extent can vary greatly across countries. However, this would only affect the results if the level of under-reporting changed significantly over time within country. Please also note that Krantz et al and Lau et al. only gave consistent conclusion on under-reporting for the US and Spain, but very different for China (least under-reporting in Lau et al.), Italy and France. In any case, please provide evidence in the manuscript on the under-reporting / change in under-reporting over time if this is still the main reason for exclusion.

2. Please reconsider if language is really a criterion for exclusion. I believe different languages were used in the included countries.
---

## [Author Response · Author response to Decision Letter 2]

13 Jan 2021

Response to Editor Comments:

Additional Editor Comments (if provided):

The Authors have clarified most of the concern. However, it would be helpful to also include these explanations especially on the exclusion criteria in the manuscript, considering the following:

1. In the response, the authors provided valid reasons for the exclusion of China and Russia, especially the large variation in latitude and hence meteorological parameters. 

In the manuscript, please state this explicitly as an exclusion criterion. 

• The reasons for both China and Russia being excluded was made explicitly on lines 119-158.

Under-reporting of infections is likely common but the extent can vary greatly across countries. However, this would only affect the results if the level of under-reporting changed significantly over time within country. 

• It would also affect the results if the level of underreporting did not change over time. This is the result of several logistical reasons that include the following:

• To approximate one scenario where the number of observed cases represent 2.3 % of the actual cases in Italy, we simply multiplied the number of cases by 2.3%--as depicted in the excel sheet shown below. [See below for the derivation of the 2.3% estimate of underreporting in Italy, taken from the data in Lau et al] (1).

o Because the lead in number of cumulative daily cases are small initially (i.e. from 1/31/20 to 2/21/20) the reduction to 2.3% results in a fraction of cases or negative log values, as shown below. Because these are not actually obtainable, the initial lead in cases would not be seen and are therefore excluded from the observation and this would change the slope of the observed vs actual log of confirmed cases. 

o This is shown in the figure below titled “Italy Tmin vs Log Confirmed Cases (Actual vs 2.3%).” In fact, the slope changes by over 30% in these two scenarios. 

• If on the other hand the reduction to observed cases being 2.3% of actual cases leads to just a shift of two identical curves save for the reduction by 97.7%--i.e. where the lead in is maintained but just delayed) then there are two further problems to contend with:

o In this scenario, the initial rise in actual cases would have been shifted to a time prior to 1/22/20 when the JHU database first began collecting data. Although the data could be found from other sources, it would not have the same methodological consistency as all of the other countries included in the analysis. This would introduce a large potential variable, whereby 50 countries all had their data obtained through the same source (JHU) but one had a separate etiology. [It should be noted, as discussed previously in our second rebuttal, we explained why the initial rate of increase in confirmed cases is key to our approach in that it allows us to isolate the SARS-CoV-2 transmission rate during a period that is predominantly influenced by temperature prior to the implementation of non-pharmaceutical interventions.]

o Also confounding would be the fact that in this scenario the shift in time to an earlier course of actual cases compared to observed, would change the temperature range over which the rise in cases appeared to have occurred. This discrepancy would affect the core of our approach and analysis. We devised a two-step process to investigate the relationship between Tmin and COVID-19 transmissibility,. In the first step, we examined linear regressions of the log of the daily cumulative COVID-19 cases versus time in sequential days (Figure 1). From this analysis we calculated the resulting regression coefficients (which we termed coefficients of time (CT) with units of log cases/day) for each country. In the second step, we investigated whether CT was associated with the median daily Tmin (Tmin_Med) in each country for the same time period that cases had been sampled. Thus, although vast underreporting of cases in Italy—even if constant over time—would not affect the CT for Italy, it would affect the appropriate Tmin_Med associated with that particular rate of transmission.

• In conclusion, vastly underreported cases in Italy would have an adverse effect on the data derived from including it in our analysis. We thus explained in our manuscript that the vast underreporting of cases in Italy was the basis of its exclusion. If the editor feels that our explanation that we provide here is essential to the readers understanding and appreciation of the manuscript, we can include in a supplementary section. 

Please also note that Krantz et al and Lau et al. only gave consistent conclusion on under-reporting for the US and Spain, but very different for China (least under-reporting in Lau et al.), Italy and France. In any case, please provide evidence in the manuscript on the under-reporting / change in under-reporting over time if this is still the main reason for exclusion.

• The Lau report used two different methods of calculating the underreporting in the countries they included in their analysis: 1) estimating total COVID-19 cases and crude case-fatality risks (cCFR) (as shown in Table 3 in the paper), and 2) adjusting numbers to aCFR of Germany and South Korea (as shown in Figure 5 in the paper) (1). 

• Here is a closer examination of the data presented from the first method taken from Table 3, with the right two columns containing additional calculations of the original data from the paper that we provided for clarity:

Countries Reported Cases Estimated Cases % Reported/Estimated Fold Increase vs Italy

China 8.10E+04 1.50E+06 5% 2

South Korea 8.30E+03 3.80E+04 22% 9

Japan 8.30E+02 1.00E+04 8% 4

Italy 2.80E+04 1.20E+06 2.3% 1

France 6.60E+03 6.80E+04 10% 4

Spain 9.20E+03 1.40E+05 7% 3

Iran 1.50E+04 4.00E+05 4% 2

States 3.50E+03 2.68E+04 13% 6

• Thus, as can be seen in the table above, Italy had the lowest % reported cases (i.e. 2.3%), compared to total estimated cases, of the 8 countries examined. This was anywhere between 1/2 to 1/9th the level of reporting of the other 7 countries included in the study. 

• Thus, Lau provides evidence of the degree of underreporting in Italy and, as instructed, we included this reference and reasoning in our manuscript on lines 116-119. 

2. Please reconsider if language is really a criterion for exclusion. I believe different languages were used in the included countries.

• All mention of language as the basis of exclusion was removed from the manuscript. 

1. Lau H, Khosrawipour T, Kocbach P, Ichii H, Bania J, Khosrawipour V. Evaluating the massive underreporting and undertesting of COVID-19 cases in multiple global epicenters. Pulmonology. 2020.

---

## [Editor Report · Decision Letter 3]

15 Jan 2021

Evidence and magnitude of the effects of meteorological changes on SARS-CoV-2 transmission

PONE-D-20-17772R3

Dear Dr. Kaplin,

We’re pleased to inform you that your manuscript has been judged scientifically suitable for publication and will be formally accepted for publication once it meets all outstanding technical requirements.

Kind regards,

Eric HY Lau, Ph.D.

Academic Editor

PLOS ONE
---

## [Editor Report · Acceptance letter]

4 Feb 2021

PONE-D-20-17772R3 

Evidence and magnitude of the effects of meteorological changes on SARS-CoV-2 transmission 

Dear Dr. Kaplin:

I'm pleased to inform you that your manuscript has been deemed suitable for publication in PLOS ONE. Congratulations! Your manuscript is now with our production department. 

Kind regards, 

on behalf of

Dr. Eric HY Lau 

Academic Editor

PLOS ONE